# Training Over a Distribution of Hyperparameters for Enhanced Performance and Adaptability on Imbalanced Classification

## Abstract

Although binary classification is a well-studied problem, training reliable classifiers under severe class imbalance remains a challenge. Recent techniques mitigate the ill effects of imbalance on training by modifying the loss functions or optimization methods. We observe that different hyperparameter values on these loss functions perform better at different recall values. We propose to exploit this fact by training one model over a distribution of hyperparameter values–instead of a single value–via Loss Conditional Training (LCT). Experiments show that training over a distribution of hyperparameters not only approximates the performance of several models but actually improves the overall performance of models on both CIFAR and real medical imaging applications, such as melanoma and diabetic retinopathy detection. Furthermore, training models with LCT is more efficient because some hyperparameter tuning can be conducted after training to meet individual needs without needing to retrain from scratch. Code will be made available upon acceptance of this paper.

## 1 Introduction

Consider a classifier that takes images of skin lesions and predicts whether they are melanoma or benign (Rotemberg et al., 2020). Such a system could be especially valuable in underdeveloped countries where expert resources for screening are scarce (Cassidy et al., 2022). The dataset for this problem, along with many other practical problems, is inherently imbalanced (*i.e.*, there are far more benign samples than melanoma samples). Furthermore, there are un-even costs associated with misclassifying the two classes because predicting a benign lesion as melanoma would result in the cost of a biopsy while predicting a melanoma lesion as benign could result in the melanoma spreading before the patient can receive appropriate treatment. Unfortunately, the exact difference in the misclassification costs may not be known *a priori* and may even change after deployment. For example, the costs may change depending on the amount of biopsy resources available or the prior may change depending on the age and condition of the patient. Thus, a good classifier for this problem should (a) have good performance across a wide range of Precision-Recall tradeoffs and (b) be able to adapt to changes in the prior or misclassification costs.

Recently proposed methods mitigate the effects of training under class imbalance by using specialized loss functions and optimization techniques, which give more weight to the minority classes and promote better generalization (Du et al., 2023a; Kini et al., 2021; Rangwani et al., 2022). While this work has led to significant improvements in the balanced accuracy in the multi-class case, less work has focused on how to adapt and maintain good accuracy across a wide range of Precision-Recall tradeoffs simultaneously.

To fill this gap, we examine several state-of-the-art methods from the class imbalance literature for binary classification problems with severe imbalance. Since these problems are binary, we analyze these methods using more detailed metrics, such as Precision-Recall curves, which plot the tradeoff between recall and precision over a range of classification thresholds.[1] This analysis reveals a consistent finding across many methods: Specifically, we observe that, on the same dataset and with

---

[1]Appendix A defines and visualizes common metrics for imbalanced binary problems.

the same method, different hyperparameter values perform better at different recalls. For example, hyperparameter value $A$ may give the best precision when the recall is 0.5, value $B$ gives the best precision when recall is 0.8, and value $C$ gives the best precision when the recall is 0.99. While this fact might *prima facie* seem only to complicate the choice of hyperparameter, it also suggests a natural question: *can we train one model that achieves the best performance over all recalls?*

To answer this question, we train a single model over a *distribution* of loss-function-hyperparameter values, instead of a single value. We do so using Loss Conditional Training (LCT), which was previously shown to approximate several similar models with one model on non-classification tasks (Dosovitskiy & Djolonga, 2020). Surprisingly, we find that on imbalanced classification problems, one model trained over a distribution of hyperparameters with LCT not only *approximates* the performance of several models trained with different hyperparameter values, but actually *improves* the performance at all recalls. This is in contrast to previous work on different applications (*e.g.*, neural image compression) which found that LCT models incurred a small performance *penalty* in exchange for better computational efficiency (Dosovitskiy & Djolonga, 2020).

We apply LCT to several existing methods for training on imbalanced data and find that it improves performance on a variety of severely imbalanced datasets including binary versions of CIFAR-10/100 (Krizhevsky et al., 2009), SIIM-ISIC Melanoma (Zawacki et al., 2020), and APTOS Diabetic Retinopathy (Karthik, 2019). Specifically, we show that when used with existing loss functions, such as Focal loss (Lin et al., 2017) and VS loss (Kini et al., 2021), LCT improves the Area Under the ROC Curve (AUC), $F_1$ score, and Brier score. We also show that LCT models are both more efficient to train and more adaptable, because some of the hyperparameter tuning can be done post-training to handle changes in the prior or misclassification costs.

In summary, our contributions are as follows.

- We show that training a single model over a distribution of hyperparameters via Loss Conditional Training (LCT) improves the performance of a variety of methods designed to address class imbalance.
- We show that LCT models are more efficient to train and more adaptable, because they can be tuned in part *after* training.
- We observe that current methods for training under class imbalance require retraining with different hyperparameters for best performance if the metrics change.
- We will publish the code upon acceptance of this paper for reproducibility and to encourage follow-up research.

## 2 RELATED WORK

Many solutions[2] have been proposed to address class imbalance, including several specialized loss functions and optimization methods (Cao et al., 2019; Rangwani et al., 2022; Buda et al., 2018; Kini et al., 2021; Shwartz-Ziv et al., 2023). Perhaps the simplest of these is to compensate for class imbalance by assigning class-dependent weights in the loss function that are inversely proportional to the frequency of the class in the training set (*e.g.*, weighted cross-entropy loss (Xie & Manski, 1989)). Another popular loss function is focal loss, which down-weights "easy" samples (*i.e.*, samples with high predictive confidence) (Lin et al., 2017).

Several more recent loss functions modulate the logits with additive and multiplicative factors before they are input to the softmax function (Cao et al., 2019; Ye et al., 2020; Menon et al., 2021). These factors aim to enforce larger margins for the minority class and/or calibrate the models. Kini et al. (2021) recognized that many of the previous loss functions for addressing class imbalance can be subsumed by a single Vector Scaling (VS) loss, which performs well on multi-class datasets after hyperparameter tuning (Kini et al., 2021). Du et al. (2023a) combine a global and local mixture consistency loss with contrastive learning and a dual head architecture.

Among work that focuses on the training method, Rangwani et al. (2022) proposed using Sharpness Aware Minimization (SAM) as an alternative optimizer that can help the model escape saddle points in multi-class problems with imbalance (Foret et al., 2021). Similarly, Shwartz-Ziv et al.

---

[2] Appendix B analyzes past literature in greater detail.

(2023) identify several training modifications—including batch size, data augmentation, specialized optimizers, and label smoothing—which can all improve performance.

All these methods have hyperparameters, which are typically tuned for best overall accuracy on a balanced test set. We observe that the choice of hyperparameters depends on the desired performance metric and propose to train one model on a range of hyperparameters using Loss Conditional Training (LCT) (Dosovitskiy & Djolonga, 2020). LCT was proposed as a method to improve efficiency by training one model that can approximate several more specific models. For example, Dosovitskiy & Djolonga (2020) propose using LCT to train one lossy image compression model which works on a range of compression rates, eliminating the need to train a separate model for each desired compression rate. They achieve greater efficiency and adaptability for image compression at a slight *cost* in performance. In contrast, we propose using LCT to *improve* the performance of a classifier under imbalance and increase its adaptability.

## 3 PRELIMINARIES

### 3.1 PROBLEM FORMALIZATION

We focus on binary classification problems with class imbalance. Specifically, let $D = \{(\mathbf{x}_i, y_i)\}_{i=1}^n$ be the training set consisting of $n$ *i.i.d.* samples from a distribution on $\mathcal{X} \times \mathcal{Y}$ where $\mathcal{X} = \mathbb{R}^d$ and $\mathcal{Y} = \{-, +\}$. Let $n_-, n_+$ be the number of negative and positive samples, respectively. We assume that $n_- > n_+$ (*i.e.*, $-$ is the majority class and $+$ is the minority class) and measure the amount of imbalance in the dataset by $\beta = n_-/n_+ > 1$.

### 3.2 PREDICTOR AND PREDICTIONS

Let $f$ be a predictor with weights $\boldsymbol{\theta}$ and $\mathbf{z} = f(\mathbf{x}, \boldsymbol{\theta})$ be $f$'s output for input $\mathbf{x}$. The entries of $\mathbf{z} = (z_-, z_+)$ are logits, *i.e.*, unnormalized scalars. Then the model's prediction is

$$\hat{y} = \begin{cases} + & \text{if } z_+ > z_- + t, \\ - & \text{otherwise,} \end{cases} \tag{1}$$

where $t \in \mathbb{R}$ is a constant and $t = 0$ by default[3]. To find either the Precision-Recall or the ROC curve of a classifier $f$, we compute the predictions over a range of $t$ values.

### 3.3 LOSS FUNCTIONS

We use our method with multiple existing loss functions, defined next.

**Focal Loss.** The focal loss down-weights samples that have high predictive confidence (*i.e.*, samples that are correctly classified and have a large margin) (Lin et al., 2017). This has the effect of reducing the amount of loss attributed to "easy examples." Let $p_y$ be the softmax score for the target class $y$, where $p_y = \frac{e^{z_y}}{\sum_{c \in \mathcal{Y}} e^{z_c}}$. Then the $\alpha$-balanced variant of focal loss is

$$\ell_{\text{Focal}}(y, \mathbf{z}) = -\alpha_y (1 - p_y)^\phi \log p_y, \tag{2}$$

where $\phi \geq 0$ is the tunable focus hyperparameter and $\alpha \in [0, 1]$ is a hyperparameter that controls the class weights such that $\alpha_+ = \alpha$ and $\alpha_- = 1 - \alpha$.

**Vector Scaling (VS) loss.** The Vector Scaling (VS) loss is designed to improve the balanced accuracy of a multi-class classifier trained on class-imbalanced data (Kini et al., 2021). This loss is a modification of the weighted cross-entropy loss with two hyperparameters $\boldsymbol{\Delta}, \boldsymbol{\iota}$ that specify an affine transformation $\Delta_c z_c + \iota_c$ for each logit $z_c$:

$$\ell_{\text{VS}}(y, \mathbf{z}) = -\log \left( \frac{e^{\Delta_y z_y + \iota_y}}{\sum_{c \in \mathcal{Y}} e^{\Delta_c z_c + \iota_c}} \right). \tag{3}$$

---

[3]The condition $z_+ > z_- + t$ is equivalent to $p_+ = \frac{e^{z_+}}{e^{z_+} + e^{z_-}} = \frac{1}{1 + e^{-(z_+ - z_-)}} > \frac{1}{1 + e^{-t}}$, so $t = 0$ is equivalent to using 0.5 as the softmax threshold.

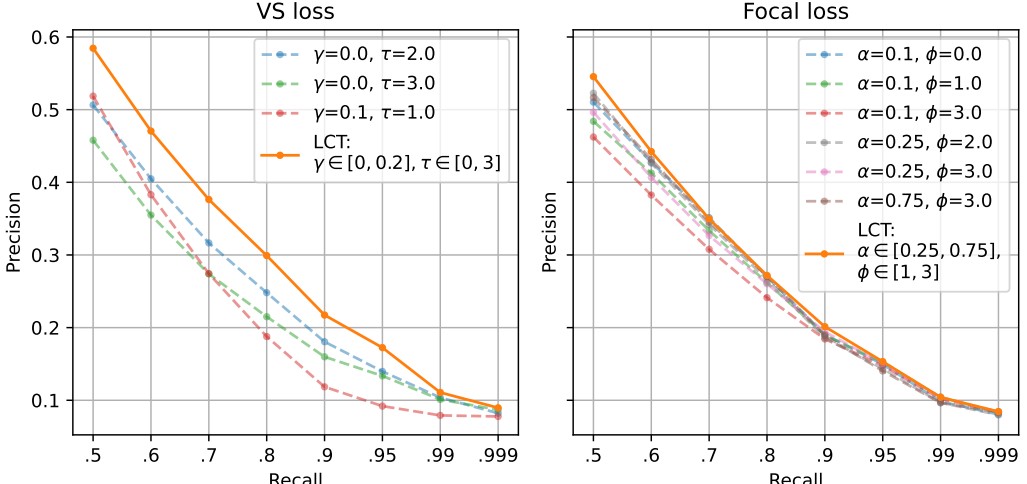

Figure 1: Effect of hyperparameters on precision and recall. Models were trained on the SIIM-ISIC Melanoma dataset with $\beta = 200$ and with the hyperparameters in Table 2. Each plot shows 1) all models with regular loss functions that achieve the highest precision at *some* recall and 2) the LCT model that achieves the best Average Precision (orange line). Results are plotted over eight different recalls and are averaged over three random initialization seeds. **With regular loss functions, different hyperparameters are better at different recall values. One model trained over a range of hyperparameters using LCT *improves* performance at all recall points.**

Like Kini et al. (2021), we use the following parameterization: $\iota_c = \tau \log\left(\frac{n_c}{n}\right)$ and $\Delta_c = \left(\frac{n_c}{n_-}\right)^{\gamma}$, where $\tau \geq 0$ and $\gamma \geq 0$ are hyperparameters set by the user. In the binary case, we can simplify the loss as follows (see Appendix E for details):

$$\ell_{\text{VS}}(-, \mathbf{z}) = \log\left(1 + e^{\eta}\right) \quad \text{and} \quad \ell_{\text{VS}}(+, \mathbf{z}) = \log\left(1 + e^{-\eta}\right), \tag{4}$$

where $\eta = \frac{z_+}{\beta^{\gamma}} - (z_- + \tau \log \beta)$. The VS loss is equivalent to the cross entropy loss when $\gamma = \tau = 0$.

### 3.4 SHARPNESS AWARE MINIMIZATION (SAM)

Sharpness Aware Minimization (SAM) is an optimization technique that aims to improve the generalization of models by finding parameters that lie in neighborhoods of uniformly low loss (Foret et al., 2021). SAM achieves this by optimizing the following objective:

$$\boldsymbol{\theta}_{\text{SAM}}^* = \arg\min_{\boldsymbol{\theta}} \max_{\|\boldsymbol{\epsilon}\|_2 \leq \rho} \mathcal{L}(\boldsymbol{\theta} + \boldsymbol{\epsilon}), \tag{5}$$

where the hyperparameter $\rho$ controls the neighborhood size and $\boldsymbol{\theta}$ are the network parameters.

## 4 TRAINING OVER A DISTRIBUTION OF HYPERPARAMETER VALUES

### 4.1 LIMITATIONS OF EXISTING METHODS

We examined several existing methods for addressing severe class imbalance in binary classification. Although the loss functions used in these methods work better than basic loss functions such as cross entropy, we notice that the best hyperparameter choice varies drastically depending on the optimization metric.

For example, in Figure 1, we train several models using both VS loss and Focal loss with different hyperparameters on the SIIM-ISIC Melanoma dataset with severe imbalance ($\beta = 200$). The hyperparameters used are shown in Table 2. For each model, we plot its precision at each of eight recall

levels. We can see that, for both VS loss and Focal loss, no single hyperparameter value works best across all recalls. For example, with VS loss, the best hyperparameters when recall=0.5 are $\gamma = 0.1, \tau = 1$, but the best hyperparameters at recall=0.8 are $\gamma = 0, \tau = 2$, and recall=0.99 are $\gamma = 0, \tau = 3$. Thus, no matter which model is chosen, performance is sacrificed at some range of the Precision-Recall curve. This also means that we will have to train new models from scratch using different hyperparameters if our Precision-Recall preference changes in real applications.

## 4.2 LOSS CONDITIONAL TRAINING (LCT)

The observation from the previous section implies that models learn different information based on the hyperparameter values of their loss function. This fact motivated us to train one model over several hyperparameter values with the aim that this model would learn to approximate the best performance of several models trained with single loss functions. We find that Loss Conditional Training (Dosovitskiy & Djolonga, 2020) is a suitable approach to implement this model. LCT was proposed as an approximation method to reduce the computational redundancy involved in training separate models with slightly different loss functions, such as neural image compression models with different compression rates. We find that we can apply it to the specialized loss functions used in the imbalance literature.

To define LCT, let $\mathcal{L}(\cdot, \cdot, \boldsymbol{\lambda})$ be a family of loss functions parameterized by $\boldsymbol{\lambda} \in \boldsymbol{\Lambda} \subseteq \mathbb{R}^{d_\lambda}$. For example, we can parameterize focal loss by defining $\boldsymbol{\lambda} = (\alpha, \phi)$ and VS loss by defining $\boldsymbol{\lambda} = (\gamma, \tau)$. A non-LCT training session finds the weights $\boldsymbol{\theta}$ of a model $f$ that minimize a loss function $\mathcal{L}(\cdot, \cdot, \boldsymbol{\lambda}_0)$ for a single combination $\boldsymbol{\lambda}_0$ of hyperparameters:

$$\boldsymbol{\theta}^*_{\boldsymbol{\lambda}_0} = \arg\min_{\boldsymbol{\theta}} \mathbb{E}_{(\mathbf{x},y) \sim D} \, \mathcal{L}(y, f(\mathbf{x}, \boldsymbol{\theta}), \boldsymbol{\lambda}_0) \, . \tag{6}$$

In contrast, LCT optimizes over a distribution $P_{\boldsymbol{\Lambda}}$ of $\boldsymbol{\lambda}$ values *and* feeds $\boldsymbol{\lambda}$ to both model and loss:

$$\boldsymbol{\theta}^* = \arg\min_{\boldsymbol{\theta}} \mathbb{E}_{\boldsymbol{\lambda} \sim P_{\boldsymbol{\Lambda}}} \mathbb{E}_{(\mathbf{x},y) \sim D} \, \mathcal{L}(y, f(\mathbf{x}, \boldsymbol{\theta}, \boldsymbol{\lambda}), \boldsymbol{\lambda}) \, . \tag{7}$$

Specifically, LCT is implemented for Deep Neural Network (DNN) predictors by augmenting the network to take an additional input vector $\boldsymbol{\lambda}$ along with each data sample $\mathbf{x}$. During training, $\boldsymbol{\lambda}$ is sampled from $P_{\boldsymbol{\Lambda}}$ for each mini-batch and is used in two ways: 1) as an additional input to the network and 2) in the loss function for the mini-batch. During inference, the model takes as input a $\boldsymbol{\lambda}$, whose value is determined by fine-tuning, along with $\mathbf{x}$, and outputs a prediction.

To condition the model on $\boldsymbol{\lambda}$, the DNN is augmented with Feature-wise Linear Modulation (FiLM) layers (Perez et al., 2018). These are small neural networks that take the conditioning parameter $\boldsymbol{\lambda}$ as input and output vectors $\boldsymbol{\mu}$ and $\boldsymbol{\sigma}$ that modulate the activations channel-wise based on the value of $\boldsymbol{\lambda}$. Specifically, consider a DNN layer with an activation map of size $H \times W \times C$. LCT transforms each activation $\mathbf{f} \in \mathbb{R}^C$ to $\tilde{\mathbf{f}} = \boldsymbol{\sigma} * \mathbf{f} + \boldsymbol{\mu}$, where both $\boldsymbol{\mu}$ and $\boldsymbol{\sigma}$ are vectors of size $C$ and "$*$" stands for element-wise multiplication.

## 4.3 DETAILS ABOUT IMPLEMENTING LCT FOR CLASSIFICATION

We let $\boldsymbol{\lambda} = (\alpha, \phi)$ for Focal loss and $\boldsymbol{\lambda} = (\gamma, \tau)$ for VS loss.[4] We draw one value of $\boldsymbol{\lambda}$ from $P_{\boldsymbol{\Lambda}}$ with each mini-batch by independently sampling each hyperparameter in $\boldsymbol{\lambda}$ from a probability density function (pdf) $L(a, b, h_b)$ over the real interval $[a, b]$. The user specifies $a, b$, and the value $h_b$ of the pdf at $b$. The value at $a$ is then found to ensure unit area. Unlike the triangular pdf, our pdf is not necessarily zero at either endpoint. Unlike the uniform pdf, it is not necessarily constant over $[a, b]$.[5] We vary $a, b, h_b$ to experiment with different distributions. The choice of $\boldsymbol{\lambda}$ during inference is determined by hyperparameter tuning, and we evaluate the model with multiple values of $\boldsymbol{\lambda}$.

In practice, tuning LCT models is significantly more efficient than tuning baseline models because much of the tuning can be done *after* training. Specifically, with the baseline models, we must train a new model for each new hyperparameter value. With LCT, we can train one network over a wide range of hyperparameter values (one $P_{\boldsymbol{\Lambda}}$ distribution) and then choose the best $\boldsymbol{\lambda}$ by simply performing inference on the validation set.

---

[4]Appendix F shows results for different choices of $\boldsymbol{\lambda}$.

[5]Appendix D contains more details about the linear distribution.

Table 1: List of training methods.

| Method | Loss | Optimizer | Parameterization of loss function |
|---|---|---|---|
| Focal | Focal | SGD/Adam | One value of $(\alpha, \phi)$ |
| Focal + LCT | Focal | SGD/Adam | Random values of $(\alpha, \phi)$ sampled from $P_{\boldsymbol{\Lambda}}$ |
| VS | VS | SGD/Adam | One value of $(\gamma, \tau)$ |
| VS + LCT | VS | SGD/Adam | Random values of $(\gamma, \tau)$ sampled from $P_{\boldsymbol{\Lambda}}$ |
| VS + SAM | VS | SAM | One value of $(\gamma, \tau)$ |
| VS + SAM + LCT | VS | SAM | Random values of $(\gamma, \tau)$ sampled from $P_{\boldsymbol{\Lambda}}$ |

Table 2: Hyperparameters used for training and evaluating each method. For each method we train 16 models, one for every combination of hyperparameters: $\boldsymbol{\lambda}$ parameters for Focal and VS, $P_{\boldsymbol{\Lambda}}$ parameters for LCT models. For LCT models, we evaluate each model with every combination of evaluation $\boldsymbol{\lambda}$ hyperparameters. For instance, for VS + LCT, where $\boldsymbol{\lambda} = (\gamma, \tau)$, we evaluate each of the $4 \times 4 = 16$ trained models on $4 \times 5 = 20$ different sets of evaluation hyperparameters. LCT pdfs for training are specified as $L(a, b, h_b)$.

| Method | Training | | Inference | |
|---|---|---|---|---|
| | $\alpha$ | $\phi$ | $\alpha$ | $\phi$ |
| Focal | 0.1, 0.25, 0.5, 0.75 | 0, 1, 2, 3 | N/A | N/A |
| Focal + LCT | $L(0.25, 0.75, 2), L(0.25, 0.75, 0),$ $L(0.25, 0.75, 4), L(0.1, 0.9, 1.25)$ | $L(1, 3, 0.5), L(1, 3, 0),$ $L(1, 3, 2), L(1, 4, 0.33)$ | 0.1, 0.25, 0.5, 0.75 | 0, 1, 2, 3 |
| | $\gamma$ | $\tau$ | $\gamma$ | $\tau$ |
| VS | 0.0, 0.1, 0.2, 0.3 | 0, 1, 2, 3 | N/A | N/A |
| VS + LCT | $L(0, 0.3, 0), L(0, 0.3, 3.3),$ $L(0, 0.2, 5), L(0.1, 0.3, 5),$ | $L(0, 3, 0), L(0, 3, 0.33),$ $L(1, 4, 0), L(1, 4, 0.33)$ | 0, 0.1, 0.2, 0.3 | 0, 1, 2, 3, 4 |

## 5 EXPERIMENTS

### 5.1 EXPERIMENTAL SETUP

**Training methods.** The training methods used in our experiment are listed in Table 1.

**Datasets.** We experiment on both toy datasets derived from CIFAR-10/100 and more realistic datasets of medical imaging applications derived from Kaggle competitions. CIFAR-10 automobile/truck and CIFAR-10 automobile/deer consists of the automobile, truck, and deer classes from the CIFAR-10 dataset. CIFAR-10 animal/vehicle consists of the 6 animal classes (bird, cat, deer, dog, frog, and horse) and the 4 vehicle classes (airplane, automobile, ship, and truck) of CIFAR-10 (Krizhevsky et al., 2009). CIFAR-100 household electronics vs. furniture was proposed as a binary dataset with imbalance by Wang et al. (2016). Each class contains 5 classes from CIFAR-100: electronics contains clock, computer keyboard, lamp, telephone and television; and furniture contains bed, chair, couch, table and wardrobe. For all CIFAR experiments, we use the provided train and test splits and subsample the minority class to obtain imbalance ratio $\beta = 100$. SIIM-ISIC Melanoma is a classification dataset designed by the Society for Imaging Informatics in Medicine (SIIM) and the International Skin Imaging Collaboration (ISIC) for a Kaggle competition (Zawacki et al., 2020). This is a binary dataset where 8.8% of the samples belong to the positive (melanoma) class (*i.e.*, $\beta = 11.4$). We follow Fang et al. (2024) and combine the 33,126 and 25,331 images from 2020 and 2019 respectively and create an 80/20 train/validation split from the aggregate data. We subsample the dataset to obtain imbalance ratios $\beta = 10, 100, 200$. Table 12 in the Appendix lists the sizes of the resulting sets. APTOS 2019 dataset was developed for a Kaggle competition organized by the Asia Pacific Tele-Ophthalmology Society (APTOS) to enhance medical screening for diabetic retinopathy (DR) in rural regions (Karthik, 2019). The 3,662 images were captured through fundus photography from multiple clinics using different cameras over an extended period. Clinicians label the images on a scale from 0 to 4, indicating the severity of DR. We convert the dataset into a binary format by designating class 0 (no DR) as the majority class,

Table 3: AUC for various datasets and training methods with and without LCT. Higher AUC indicates better performance. We train and evaluate over the hyperparameters in Table 2 and report the best values for each method. The imbalance ratio $\beta$ equals 100 for CIFAR experiments. For each dataset, we bold the value corresponding to the best method and underline the value of the better method between the baseline and LCT. **LCT improves the AUC, especially on the Melanoma dataset at high imbalance ratios.**

| Method | CIFAR-10/100 | | | | SIIM-ISIC Melanoma | | | APTOS | |
|---|---|---|---|---|---|---|---|---|---|
| | Auto Deer | Auto Truck | Animal Vehicle | House-hold | $\beta$=10 | $\beta$=100 | $\beta$=200 | $\beta$=100 | $\beta$=200 |
| Focal | 0.985 | 0.929 | 0.982 | 0.699 | 0.951 | 0.905 | 0.891 | 0.985 | 0.982 |
| Focal + LCT | 0.985 | 0.927 | 0.981 | **0.714** | 0.951 | 0.910 | 0.902 | **0.987** | **0.984** |
| VS | 0.980 | 0.918 | 0.981 | 0.710 | **0.954** | 0.905 | 0.884 | 0.980 | 0.980 |
| VS + LCT | 0.983 | 0.930 | 0.983 | 0.705 | **0.954** | **0.914** | **0.911** | 0.983 | **0.984** |
| VS + SAM | 0.985 | 0.923 | **0.988** | 0.709 | 0.938 | 0.895 | 0.892 | 0.613 | 0.582 |
| VS + SAM + LCT | **0.988** | **0.934** | 0.987 | 0.708 | 0.893 | 0.650 | 0.831 | 0.715 | 0.622 |

while combining the other classes, representing various degrees of DR severity, into one minority class. We divide the data in the ratio 80/20 for train/validation following Fang et al. (2024) and subsample the minority class to $\beta = 200$.

**Hyperparameters.** For each dataset, imbalance ratio $\beta$, and method, we train 16 models with different hyperparameters. Specifically, for baseline models, that is, models without LCT, hyperparameter combinations are $\boldsymbol{\lambda}$ vectors. For LCT models, hyperparameter combinations determine the distribution $P_{\boldsymbol{\Lambda}}$ from which $\boldsymbol{\lambda}$ is drawn. We also evaluate LCT models with different combinations $\boldsymbol{\lambda}$ of hyperparameters. Table 2 contains more details. For each set of hyperparameters, we average results over three random initialization seeds for training. These seeds affect both the initial model weights and the shuffling of the training data loader.

**Model architectures and training procedure.** We use three model architectures and training procedures, following the literature for each type of dataset. For CIFAR-10/100 data, we use a ResNet-32 model architecture with 64 channels in the first layer (He et al., 2016). We train these models from scratch for 500 epochs, using an initial learning rate of 0.1 and decreasing this to $10^{-3}$ and $10^{-5}$ at 400 and 450 epochs respectively. For the Melanoma dataset, we follow Shwartz-Ziv et al. (2023) and use ResNext50-32x4d (He et al., 2016) pre-trained on ImageNet (Deng et al., 2009). We train the Melanoma models for 10 epochs following the learning rates and weight decays of Fang et al. (2024). For APTOS models, we employ ConvNext-tiny (Liu et al., 2022) pre-trained on ImageNet (Deng et al., 2009). We modify the settings in Fang et al. (2024) a bit, since our dataset is different from theirs (binary instead of multi-class). We train models for 35 epochs, with a learning rate of $5 \times 10^{-5}$, no weight decay, and the cosine scheduler.

To apply LCT, we augment the networks with one FiLM block (Perez et al., 2018) after the final convolutional layer and before the linear layer. This block consists of two linear layers with 128 hidden units. Specifically, the first layer takes one input $\boldsymbol{\lambda}$ and outputs 128 hidden values, and the second layer outputs 64 values which are used to modulate the 64 channels of convolutional activations (total of 8448 additional parameters for two-dimensional $\boldsymbol{\lambda}$).

For all experiments, we use a batch size of 128 and gradient clipping with maximum norm equal to 0.5. For most datasets and models, we use Stochastic Gradient Descent (SGD) optimization with momentum $= 0.9$. For APTOS datasets, we use Adam (Kingma, 2014), following Fang et al. (2024). Additionally, for models with SAM optimization, we use $\rho = 0.1$. Note that we can train a model with both LCT and SAM because these affect different parts of the training procedure. Specifically, while LCT affects the model's inputs and the calculation of the loss, SAM affects how the model uses the information from the loss to update its weights.

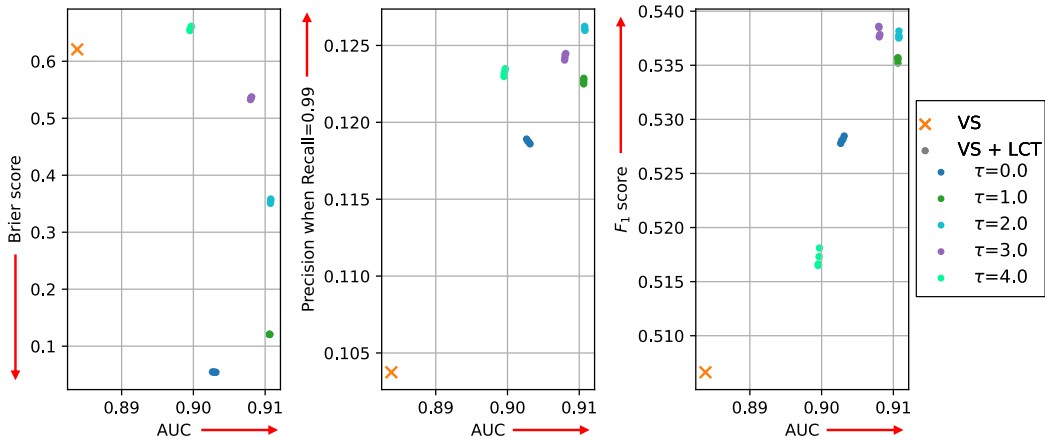

Figure 2: AUC versus Brier score (left), precision when recall is $\geq 0.99$ (center) and $F_1$ score (right) obtained from training one VS model (orange cross) and one VS + LCT model (dots) on the SIIM-ISIC Melanoma dataset with $\beta = 200$. For both methods, we show results for the model that obtains the highest AUC. For the LCT model, we evaluate with the 20 different $\boldsymbol{\lambda}$ values from Table 2 and color the dots by the value of $\tau$ as in the legend. **One LCT model can be adapted *after* training to optimize a variety of metrics.**

## 5.2 OVERALL PERFORMANCE

**Precision-Recall curves.** We first consider how training over a range of hyperparameters via LCT affects performance over a wide range of classification thresholds. Figure 1 compares the baseline models to the LCT models over eight recall values on the Precision-Recall curves. This figure shows that training over a distribution of hyperparameters with LCT yields a *single* model that *improves* the precision over the best baseline models at all recall values. This is in contrast to the original applications of LCT, where LCT is only designed to approximate (but not improve) multiple models (Dosovitskiy & Djolonga, 2020).

**AUC.** We next consider more comprehensive results using the Area Under the Receiver Operating Characteristic (ROC) curves (AUC) (Egan, 1975). There is a one-to-one correspondence between ROC curves and Precision-Recall curves (Davis & Goadrich, 2006). We include AUC results in the main body because it has meaningful interpretations (Flach & Kull, 2015) and include results for Average Precision (which summarizes Precision-Recall curves) in Table 13 of the Appendix. These curves both show model performance over a wide range of classification thresholds, making AUC and Average Precision good metrics for the situations we discussed in the introduction.

Table 3 shows AUC results for various methods designed to address class imbalance and various binary datasets with severe class imbalance. LCT consistently improves the VS and Focal methods; however, it has inconsistent performance on the VS + SAM method. Furthermore, for most datasets (all except CIFAR-10 animal/vehicle), the best performing method is an LCT method. This shows that models trained with LCT are generally better at ranking the predictions (*i.e.*, giving positive samples higher probabilities of being positive than negative samples). This makes sense because, like baseline models, LCT always penalizes incorrect *relative* rankings within the same batch; however, unlike baseline models, the model does not overfit to one *absolute* bias (associated with one single hyperparameter), but rather a random bias drawn with each batch. This property also makes LCT easily adaptable to different Precision-Recall preferences.

Table 4: AUC for baseline, LCT, and LCT without FiLM layers for CIFAR-10 Automobile/Truck with $\beta = 100$. Bold values are the best in each column. **LCT's improvement is not just from adding randomness to the loss, but also from the additional information conveyed by $\lambda$.**

| Method | VS | VS + SAM | Focal |
|---|---|---|---|
| Baseline | 0.918 | 0.923 | **0.929** |
| +LCT | **0.930** | **0.934** | 0.927 |
| +LCT without FiLM | 0.886 | 0.908 | 0.926 |

### 5.3 MODEL ADAPTABILITY

Since LCT models are trained over a distribution $P_{\Lambda}$ of $\lambda$ values, they can be evaluated over any $\lambda$ value within (or maybe even outside of) $P_{\Lambda}$. In this section, we show how this feature leads to adaptable models, or models that can be further tuned *after* training.

Figure 2 compares the results obtained by training one baseline (VS) model and one LCT (VS + LCT) model. While the baseline model has only one output, the LCT model has many potential outputs based on the value of the inference-time value of $\lambda$. For example, on the left plot, we see that varying the inference-time $\lambda$ can have a big effect on the Brier score of the model. Specifically, decreasing the value of $\tau$ in the $\lambda$ vector significantly decreases the Brier score (*i.e.*, improves calibration)[6], albeit at some cost in AUC in the most extreme cases. Similarly, varying the value of $\lambda$ can vary the precision for 0.99 recall (center) or the $F_1$ score (right).

Thus, training models with LCT allows a practitioner to train one good model and then efficiently adapt the model to meet specific requirements (*e.g.*, improve calibration, precision at high recalls, or $F_1$ score) after training. This property is particularly valuable for medical applications, as a single trained model can be shared by health providers across different regions, allowing them to adapt it to their specific needs despite variations in medical resources and preferences.

### 5.4 ANALYZING THE IMPACT OF LCT

In this section, we analyze whether the improvement from training classification models with LCT is a result of adding more randomness to the training procedure and/or the additional information (the added $\lambda$ input) that the model receives about the loss function. In other words, is training over $P_{\Lambda}$ instead of a fixed $\lambda$ simply a form of regularization that helps the model to generalize better, or does the additional input $\lambda$ also help the model learn to optimize over different precision-recall tradeoffs?

To test this, we train models by feeding $\lambda$ only to the loss function but not to the model. In other words, in contrast to Equation (7), we optimize

$$\theta^* = \arg\min_{\theta} \mathbb{E}_{\lambda \sim P_{\Lambda}} \mathbb{E}_{(\mathbf{x},y) \sim D} \, \mathcal{L}(y, f(\mathbf{x}, \theta), \lambda) \,. \tag{8}$$

Specifically, we draw $\lambda \sim P_{\Lambda}$ with each mini-batch during training and feed it to the loss function like we did in LCT. However, unlike LCT, we use the model architecture without FiLM layers. For inference, the model only takes $\mathbf{x}$ as input, and not $\lambda$. We call this method **LCT without FiLM**.

Table 4 shows that LCT without FiLM performs significantly worse than both the baseline and the LCT methods. Specifically, the AUC decreases from 0.918 with VS (or 0.930 with VS+LCT) to 0.886 with VS+LCT without FiLM. This shows that blindly training over a distribution of hyperparameters does not improve the generalization of the models and in fact harms the performance. However, training over a distribution of hyperparameters *and* feeding information about those hyperparameters to the model does improve overall performance in this setting.

## 6 CONCLUSION

We propose a new regimen for training binary classification models under severe imbalance. Specifically, we propose training a single model over a distribution of hyperparameters using Loss Con-

---

[6]See Appendix A for more details about the Brier score.

ditional Training (LCT). We find that this consistently improves Precision-recall curves and various other metrics. This improvement comes from the fact that training over a range of hyperparameters is a proxy for optimizing along different precision-recall tradeoffs. Furthermore, using LCT is more efficient because some hyperparameter tuning can be done after training and one model is adaptable to many circumstances. Areas of future work include adapting this method to work on multi-class classification problems under imbalance and on regression tasks.

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

# A  BINARY CLASSIFICATION METRICS

In the binary case, we assume that the minority class is the positive class (*i.e.*, the class with label a label of one). For a given classifier, we can categorize the samples in terms of their actual labels and the classifier's predictions as shown in Table 5. The remainder of this section defines several metrics used for binary classification.

|  | **Predicted** $+$ | **Predicted** $-$ |
|---|---|---|
| **Actually** $+$ $(n_+)$ | True Positives (tp) | False Negatives (fn) |
| **Actually** $-$ $(n_-)$ | False Positives (fp) | True Negatives (tn) |

Table 5: Categorization of samples in the binary case based on their actual labels (rows) and predicted labels (columns).

## A.1  TRUE POSITIVE RATE (TPR) = MINORITY-CLASS ACCURACY = RECALL

The True Positive Rate (TPR) is defined as the proportion of actual positive samples which are predicted to be positive. Note that in the binary case, this is equivalent to both the minority-class accuracy and the recall.

$$\text{TPR} = \text{Minority-class acc.} = \text{Recall} = \frac{tp}{tp + fn} = \frac{tp}{n_+} \tag{9}$$

|  | **Predicted** $+$ | **Predicted** $-$ |
|---|---|---|
| **Actually** $+$ $(n_+)$ | True Positives (tp) | False Negatives (fn) |
| **Actually** $-$ $(n_-)$ | False Positives (fp) | True Negatives (tn) |

Table 6: Visualization of TPR calculation. All shaded cells are included in calculation of metric. The numerator is highlighted in dark blue.

## A.2  FALSE POSITIVE RATE (FPR) = 1 - MAJORITY-CLASS ACCURACY = 1 - TNR

The False Positive Rate (FPR) is defined as the proportion of actual negative samples which are predicted to be positive. Note that in the binary case, this is equivalent to 1 - the majority-class and 1 - the True Negative Rate (TNR).

$$\text{FPR} = 1 - \text{Majority-class acc.} = \frac{fp}{tn + fp} = \frac{fp}{n_-} \tag{10}$$

|  | **Predicted** $+$ | **Predicted** $-$ |
|---|---|---|
| **Actually** $+$ $(n_+)$ | True Positives (tp) | False Negatives (fn) |
| **Actually** $-$ $(n_-)$ | False Positives (fp) | True Negatives (tn) |

Table 7: Visualization of FPR metric. All shaded cells are included in calculation of metric. The numerator is highlighted in dark blue.

## A.3  PRECISION

The precision is defined as the proportion of predicted positive samples which are actually positive.

$$\text{Precision} = \frac{tp}{tp + fp} \tag{11}$$

| | Predicted $+$ | Predicted $-$ |
|---|---|---|
| **Actually** $+ (n_+)$ | True Positives (tp) | False Negatives (fn) |
| **Actually** $- (n_-)$ | False Positives (fp) | True Negatives (tn) |

Table 8: Visualization of precision metric. All shaded cells are included in calculation of metric. The numerator is highlighted in dark blue.

## A.4 OVERALL ACCURACY

Perhaps the simplest metric is the overall accuracy of the classifier. This is simply the proportion of samples which are correctly classified (regardless of their class). If the test set is imbalanced, a trivial classifier which predicts all samples as negative will achieve a high overall accuracy. Specifically, the overall accuracy of this classifier will be the proportion of negative samples or $\frac{\beta}{1+\beta}$. In class imbalance literature, the overall accuracy is often reported on a balanced test set. In this case, the accuracy is an average accuracy on the positive and negative classes.

$$\text{Overall accuracy} = \frac{tp + tn}{tp + fn + fp + tn} = \frac{tp + tn}{n} \tag{12}$$

| | Predicted $+$ | Predicted $-$ |
|---|---|---|
| **Actually** $+ (n_+)$ | True Positives (tp) | False Negatives (fn) |
| **Actually** $- (n_-)$ | False Positives (fp) | True Negatives (tn) |

Table 9: Visualization of overall accuracy metric. All shaded cells are included in calculation of metric. The numerator is highlighted in dark blue.

## A.5 BALANCED ACCURACY

While overall accuracy is the accuracy across the whole test set, balanced accuracy is the average accuracy of the $+$ and $-$ samples. If the test set is balanced (*i.e.*, has $\beta = 1$), then these two accuracies are equal.

$$\text{Balanced acc.} = 0.5 \left( \frac{tp}{n_+} + \frac{tn}{n_-} \right) \tag{13}$$

## A.6 F-SCORES

In some problems, such as information retrieval, there is only one class of interest (the positive class) and the true negatives can vastly outnumber the other three categories. In this case, a method's effectiveness is determined by 1) how many positive samples it correctly predicted as positive (*i.e.*, the recall) and 2) how many samples are actually positive out of all the samples it predicted as positive (*i.e.*, the precision). The $F_1$ metric measures how well a method can achieve both of these goals simultaneously. Specifically, the $F_1$ measure is the harmonic mean between the precision and recall and is defined as

$$F_1 = \frac{2 \cdot \text{precision} \cdot \text{recall}}{\text{precision} + \text{recall}}. \tag{14}$$

The $F_1$ measure assumes that the precision and recall have equal weights; however, sometimes problems have different costs for recall and precision. These asymmetric costs can be addressed by the more general $F_\beta$ metric. Let $\beta$ be the ratio of importance between recall and precision, then $F_\beta$ is defined as [7],

$$F_\beta = \frac{(1 + \beta^2) \cdot \text{precision} \cdot \text{recall}}{\beta^2 \cdot \text{precision} + \text{recall}}. \tag{15}$$

---

[7]Note that this $\beta$ differs from the $\beta$ which we defined in the main body.

|  | Predicted $+$ | Predicted $-$ |
|---|---|---|
| Actually $+$ ($n_+$) | True Positives (tp) | False Negatives (fn) |
| Actually $-$ ($n_-$) | False Positives (fp) | True Negatives (tn) |

Table 10: Visualization of $F_\beta$ metric. All shaded cells are included in calculation of metric. The tp cell is highlighted in dark blue because it is included in both the precision and recall calculation.

### A.7 G-MEAN

The Geometric mean (G-mean or GM) is the geometric mean of the TPR (*i.e.*, sensitivity) and TNR (*i.e.*, specificity) and is defined as follows,

$$GM = \sqrt{\text{SE} * \text{SP}} \tag{16}$$

$$= \sqrt{\frac{\text{tp}}{\text{tp} + \text{fn}} * \frac{\text{tn}}{\text{tn} + \text{fp}}}. \tag{17}$$

|  | Predicted $+$ | Predicted $-$ |
|---|---|---|
| Actually $+$ ($n_+$) | True Positives (tp) | False Negatives (fn) |
| Actually $-$ ($n_-$) | False Positives (fp) | True Negatives (tn) |

Table 11: Visualization of G-mean metric. All shaded cells are included in calculation of metric. Values in the numerator are highlighted in dark blue.

### A.8 ROC CURVES AND AUC

Of course, the number of true positives and true negatives are a trade-off and any method can be modified to give a different combination of these metrics. Specifically, the decision threshold can be modified to give any particular recall. Receiver Operating Characteristic (ROC) curves take this in consideration and show the trade-off of true positive rates and false positive rates over all possible decision thresholds. The area under the ROC curve (AUC) is calculated using the trapezoid rule for integration over a sample of values and is a commonly used metric (Buda et al., 2018).

### A.9 PRECISION-RECALL CURVES AND AVERAGE PRECISION

Similarly, the precision-recall curves show the tradeoff of the recall (on the x-axis) and the precision (on the y-axis) over all possible thresholds. The Average Precision (AP) summarizes the precision-recall curve. To define it let $P_t, R_t$ be the precision and recall at threshold $t$. Then

$$AP = (R_n - R_{n-1})P_n. \tag{18}$$

### A.10 BRIER SCORE

The Brier score is often used to measure the calibration of the model or how well the model's predicted probabilities match the true likelihood of outcomes (Brier, 1950). For example, with a well-calibrated model, 80% of samples with predicted probability 0.8 will be positive. Good calibration allows for optimal decision thresholds that can be adjusted easily for different priors (Kull et al., 2017; Bella et al., 2010).

The Brier score (equivalently the Mean Squared Error), is defined as

$$BS = \sum_{i=1}^{N}(y_i - p_i)^2, \tag{19}$$

where $y_i$ is the true sample label (0 or 1) and $p_i$ is the predicted probability of the sample being 1 (*e.g.*, the softmax score for class 1). Note that *lower* Brier scores indicate better calibration.

## B    Detailed related work

Training on imbalanced datasets with algorithms designed to work on balanced datasets can be problematic because the gradients and losses are biased towards the common classes, so the rare classes will not be learned well. Current methods to mitigate the effects of training under imbalance include methods that modify the loss functions, re-sample and augment training samples, and improve the module via two-stage learning, ensembles, or representation learning Zhang et al. (2023).

### B.1    Specialized loss functions

To balance the gradients from all classes, several papers adaptively change a sample's weight in the loss based on features such as the sample's confidence score, class frequency, and influence on model weights Zhang et al. (2021); Fernando & Tsokos (2021); Park et al. (2021); Wang et al. (2021a); Li et al. (2021a). Other work addresses the difference in the norms of features associated with frequent (head) and rare (tail) classes and proposes to balance this by utilizing a feature-based loss Li et al. (2022b) or weight-decay and gradient clipping Alshammari et al. (2022).

Some work has focused on enforcing larger margins on the tail classes using additive factors on the logits in Cross-entropy loss Cao et al. (2019); Menon et al. (2021); Li et al. (2022c). Other work proposed adding multiplicative factors to the logits to adjust for the difference in the magnitude of minority-class logits at training and test time or minimize a margin-based generalization bound Ye et al. (2020); Kang et al. (2021). Kini et al. (2021) show that multiplicative factors are essential for the terminal phase of training, but that these have negative effects early during training, so additive factors are necessary to speed up convergence. They propose Vector Scaling (VS) loss as a general loss function that includes both additive and multiplicative factors on the logits. Behnia et al. (2023) study the implicit geometry of classifiers trained on a special case of VS loss.

Beyond classification, many papers have focused on imbalance in instance segmentation and object detection applications Tan et al. (2021); Wang et al. (2021c); Feng et al. (2021); Li et al. (2020). Ren et al. (2022) also propose a balanced Mean Square Error (MSE) loss for regression problems, such as age and depth estimation.

### B.2    Data-level methods

Another way to balance the gradients of classes during training is resampling. This could be done by sampling the minority class more often (random over-sampling) or sampling the majority class less often (random under-sampling) Johnson & Khoshgoftaar (2019); Japkowicz (2000); Liu et al. (2009). Jiang et al. (2023); Hou et al. (2023) use clustering to drive resampling; specifically, they cluster head classes into multiple clusters and then resample across clusters instead of classes. Meta-learning has been used to estimate the optimal sampling rates of different classes Ren et al. (2020), while Wang et al. (2019) dynamically adapts both the loss function and sampling procedure throughout training.

Additionally, data augmentation can be used alongside oversampling to increase the size of the minority class samples and enable model generalization Zhong et al. (2021); Zang et al. (2021); Li et al. (2021b); Du et al. (2023a). With tabular data, small perturbations of random noise can be added to generate new examples. Images lend themselves to more high-level augmentations. Methods that copy and paste patches of images, such as CutMix Yun et al. (2019) and PuzzleMix Kim et al. (2020) have been used to improve classification or instance segmentation Ghiasi et al. (2021) performance.

SMOTE (Synthetic Minority Over-sampling Technique) creates synthetic examples by interpolating between samples in the same class of the training set Chawla et al. (2002). The interpolation is done in feature space instead of data space. The mixup method also generates synthetic examples; however, unlike SMOTE, it interpolates between samples of different classes Zhang et al. (2018). These synthetic examples are given a soft label that corresponds to the proportion of input from each class. StyleMix adapts mixup to separately manipulate an image's content and style Hong et al. (2021). ReMix advances mixup to optimize for situations with class imbalance by combining mixup and resampling Bellinger et al. (2020). Li et al. (2021b) creates augmented data samples by translating deep features along semantically meaningful directions. Zada et al. (2022) adds pure

noise images to the training set and introduces a new type of distribution-aware batch normalization layer to facilitate training with them.

### B.3 MODULE IMPROVEMENTS

Some work has shown that overly emphasizing the minority class can disrupt the original data distribution and cause overfitting Zhou et al. (2020); Du et al. (2023a) or reduce the accuracy of the majority class Zhu et al. (2022). To mitigate this issue, contrastive learning Du et al. (2023a); Zhu et al. (2022); Wang et al. (2021b) has been used to learn a robust feature representation by enforcing consistency between two differently augmented versions of the data. Du et al. (2023a) devise a single-stage approach, by combining the ideas of contrastive representations with soft label class re-weighting to achieve state-of-the-art performance on several benchmarks.

Multi-expert methods have also been explored Zhang et al. (2022); Wang et al. (2020b); Li et al. (2022a); Sanchez Aimar et al. (2023). Li et al. (2022a) collaboratively learn multiple experts, by learning individual experts and transferring knowledge among them, in a nested way. Similarly, Sanchez Aimar et al. (2023) propose using a an ensemble of experts calibrated with different logit adjustments. Methods have also used two branches or heads with one classification branch along with another branch, either to re-balance Guo & Wang (2021); Zhou et al. (2020) or calibrate Wang et al. (2020a) the model. Tang et al. (2022) adopt a generalized approach to learn attribute-invariant features that first discovers sets of samples with diverse intra-class distributions from the low confidence predictions, and then learns invariant features across them. Dong et al. (2022) uses language prompts to improve both general and group-specific features. Xu et al. (2022) propose using Dynamic Label Space Adjustments (DLSA) to first separate head and tail classes to create a series of more balanced classification tasks for one longtailed multi-class classification problems. Similarly, SuperDisco proposes using hierarchical relationships between classes and graph learning to improve the performance on longtailed multi-class classification problems (Du et al., 2023b).

## C NUMBER OF SAMPLES

| Dataset | Train set | | | | Test set | |
|---|---|---|---|---|---|---|
| | # maj. | # min. ($\beta = 10$) | # min. ($\beta = 100$) | # min. ($\beta = 200$) | # maj. | # min. |
| CIFAR10 pair | 5,000 | 500 | 50 | 25 | 1,000 | 1,000 |
| Household | 2,500 | 250 | 25 | 13 | 500 | 500 |
| Dogs vs. Cats | 11,500 | 1,500 | 150 | 75 | 1,000 | 1,000 |
| SIIM-ISIC Melanoma | 41,051 | 4,105 | 410 | 205 | 12,300 | 1,035 |
| APTOS Diabetic Retinopathy | 1,444 | 144 | 14 | 7 | 361 | 370 |

Table 12: Number of samples in majority class and minority class of the datasets at different imbalance ratios $\beta$. $\beta$ only affects the number of minority-class samples during training.

## D LINEAR PROBABILITY DENSITY FUNCTION

We use a linear probability distribution to sample $\lambda$ from an interval $[a, b]$. Unlike the triangular distribution, the probability distribution function (PDF) of this distribution can be nonzero at both a and b. Figure 3 shows several examples of this distribution when $[a, b] = [0, 3.0]$.

To implement this distribution, the user first selects the domain $[a, b]$ and the height of PDF at $b$, $h_b$. The function then calculates $h_a$ so that the area under the PDF equals one. To sample from this distribution, we draw from a uniform(0,1) distribution and use the inverse cumulative distribution function (CDF) of the linear distribution to find a value of $\lambda$.

Note that this is the uniform distribution on $[a, b]$ when $h_a = h_b$. Additionally, this is a triangular distribution when $h_a = 0$ or $h_b = 0$.

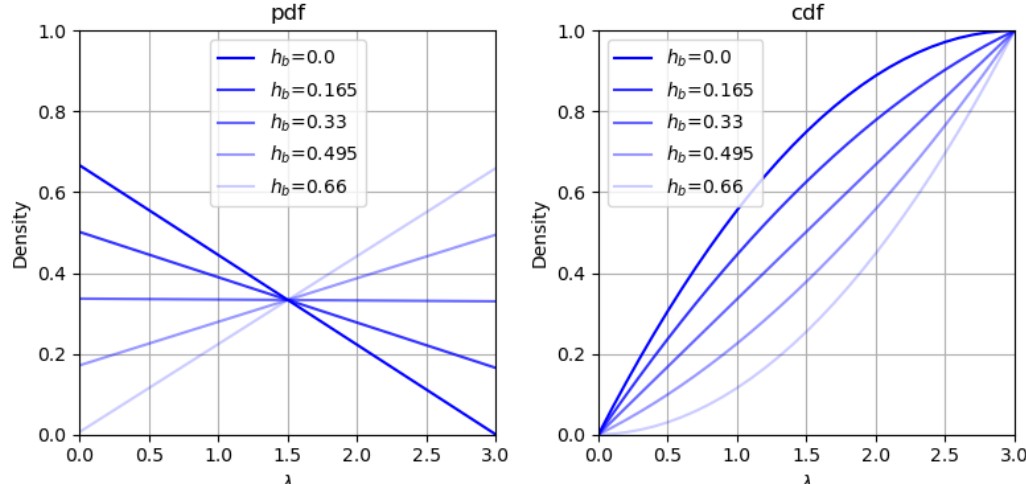

Figure 3: Example PDFs and CDFs of linear distribution with different $h_b$ when $[a, b] = [0, 3]$.

## E SIMPLIFYING VS LOSS

Recall the definition of VS loss.

$$\ell_{VS}(y, \mathbf{z}) = -\log\left(\frac{e^{\Delta_y z_y + \iota_y}}{\sum_{c \in \mathcal{Y}} e^{\Delta_c z_c + \iota_c}}\right) \tag{20}$$

Additionally, recall that $\Delta$ and $\iota$ are parameterized by $\gamma$ and $\tau$ as follows.

$$\Delta_c = \left(\frac{n_c}{n_{max}}\right)^{\gamma} \tag{21}$$

$$\iota_c = \tau \log\left(\frac{n_c}{\sum_{c' \in C} n_{c'}}\right) \tag{22}$$

Consider a binary problem with $\mathcal{Y} = \{-, +\}$ and an imbalance ratio $\beta$. Then $\Delta$'s and $\iota$'s are defined as follows

$$\Delta_- = 1 \qquad\qquad \iota_- = \tau \log\left(\frac{\beta}{\beta + 1}\right) \tag{23}$$

$$\Delta_+ = \frac{1}{\beta^{\gamma}} \qquad\qquad \iota_+ = \tau \log\left(\frac{1}{\beta + 1}\right) \tag{24}$$

$$\tag{25}$$

We can simplify $\ell_{VS}(-, \mathbf{z})$ as follows. First plug in the values of $\mathbf{\Delta}$ and $\boldsymbol{\iota}$:

$$\ell_{VS}(-, \mathbf{z}) = -\log\left(\frac{e^{z_- + \tau \log(\frac{\beta}{\beta+1})}}{e^{z_- + \tau \log(\frac{\beta}{\beta+1})} + e^{\frac{z_+}{\beta^{\gamma}} + \tau \log(\frac{1}{\beta+1})}}\right) \tag{26}$$

Then rewrite $\tau \log(\frac{a}{b})$ as $\tau \log(a) - \tau \log(b)$ and cancel out $e^{\tau \log(\beta+1)}$ from the numerator and denominator.

$$\ell_{VS}(-, \mathbf{z}) = -\log \left( \frac{e^{z_- + \tau \log(\beta) - \tau \log(\beta+1)}}{e^{z_- + \tau \log(\beta) - \tau \log(\beta+1)} + e^{\frac{z_+}{\beta^\gamma} + \tau \log(1) - \tau \log(\beta+1)}} \right) \tag{27}$$

$$= -\log \left( \frac{\frac{e^{z_- + \tau \log(\beta)}}{e^{\tau \log(\beta+1)}}}{\frac{e^{z_- + \tau \log(\beta)}}{e^{\tau \log(\beta+1)}} + \frac{e^{z_+ / \beta^\gamma}}{e^{\tau \log(\beta+1)}}} \right) \tag{28}$$

$$= -\log \left( \frac{e^{z_- + \tau \log(\beta)}}{e^{z_- + \tau \log(\beta)} + e^{z_+ / \beta^\gamma}} \right) \tag{29}$$

Then use the following two facts to a) simplify the term inside the log and b) rewrite the log term:

$$\text{a) } \frac{e^a}{e^a + e^b} = \frac{e^a}{e^a + e^b} \frac{\frac{1}{e^a}}{\frac{1}{e^a}} = \frac{1}{1 + e^{b-a}} \tag{30}$$

$$\text{b) } \log \left( \frac{1}{1 + e^{b-a}} \right) = \log(1) - \log(1 + e^{b-a}) = -\log(1 + e^{b-a}) \tag{31}$$

$$\ell_{VS}(-, \mathbf{z}) = \log \left( 1 + e^{z_+ / \beta^\gamma - z_- - \tau \log \beta} \right) \tag{32}$$

We follow a similar process for $\ell_{VS}(+, \mathbf{z})$ and get

$$\ell_{VS}(+, \mathbf{z}) = \log \left( 1 + e^{-z_+ / \beta^\gamma + z_- + \tau \log \beta} \right). \tag{33}$$

## F   METRIC HEATMAPS

In this section, we show how the choice of $\mathbf{\Lambda}$ and hyperparameters affect the AUROC. All results in this section are on the Melanoma test set and come from models trained that were trained on the Melanoma train set with $\beta = 200$. Each box (representing a unique hyperparameter choice) is the average of three models initialized with different seeds.

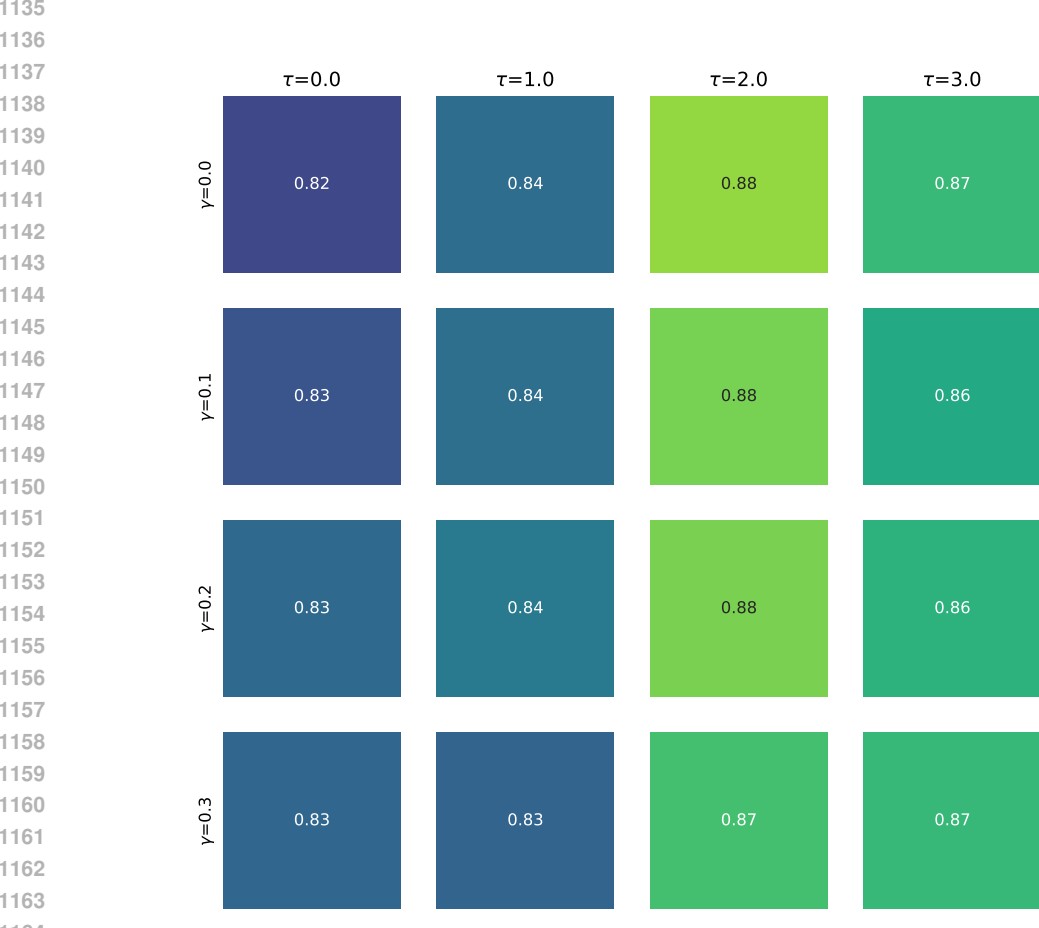

Figure 4: AUROC for 16 models trained with VS + SGD. The hyperparameters $\gamma = 0.0, \tau = 2.0$ give the best average AUROC=0.884.

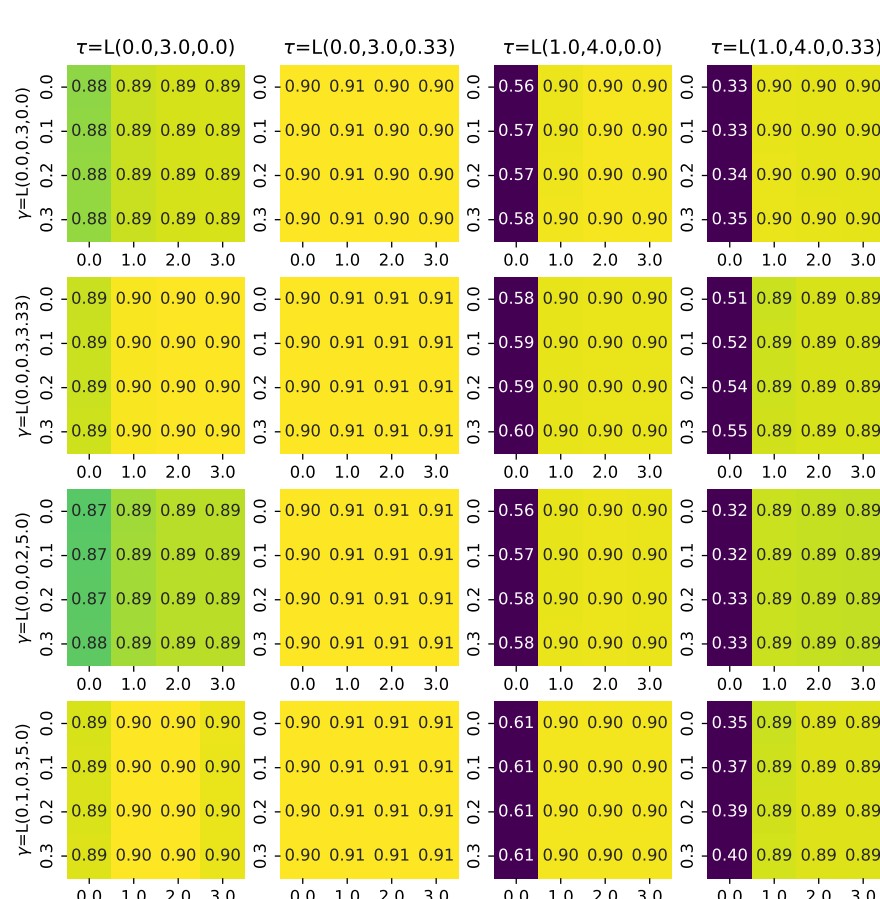

Figure 5: AUROC for 16 models trained with VS loss + LCT ($\boldsymbol{\lambda} = (\gamma, \tau)$). Each model was evaluated with 20 different hyperparameters (represented by the 20 small boxes inside of each large box). The models trained with $\gamma$ drawn from Linear(0.1, 0.3, 5.0) and $\tau$ drawn from Linear(0, 3, 0.33) and evaluated with $\gamma = 0$, $\tau = 2$ give the best AUROC=0.911. Most models outperform the best model trained with VS loss (Figure 4).

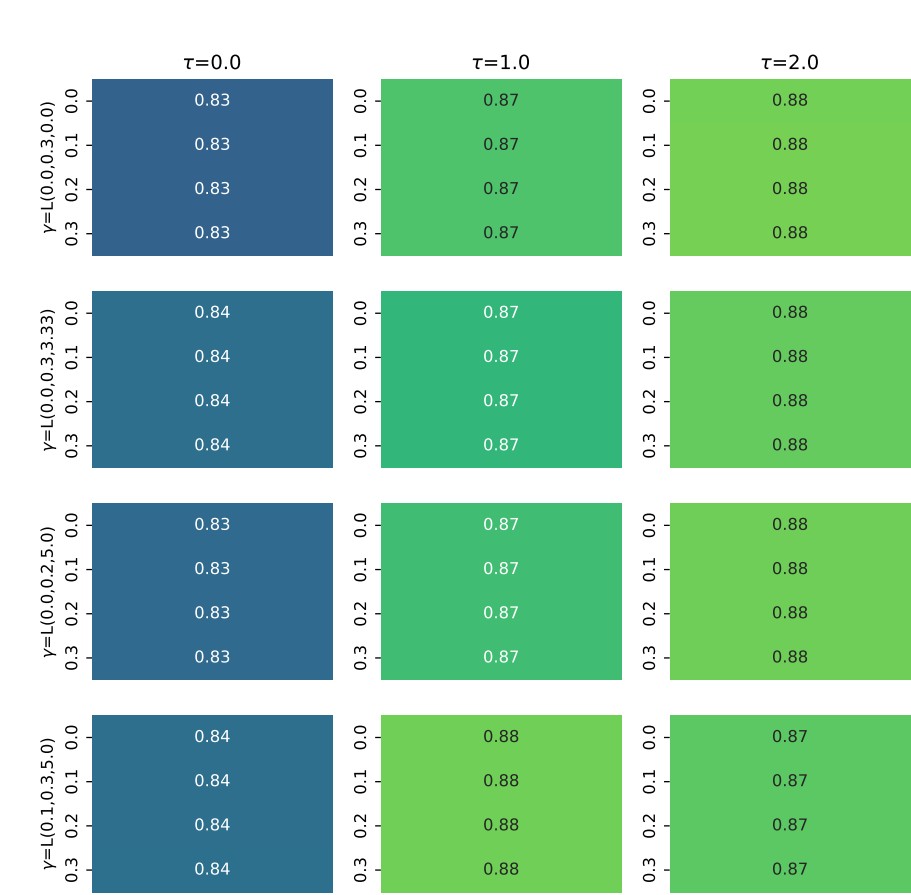

Figure 6: AUROC for 16 models trained with VS + LCT ($\boldsymbol{\lambda} = \gamma$). Each model was evaluated with 4 different hyperparameters (represented by the 4 small boxes inside of each large box). The models trained with $\tau = 2$ and $\gamma$ drawn from Linear(0.0, 0.3, 2.0) and evaluated at 0.3 give the best AUROC=0.879.

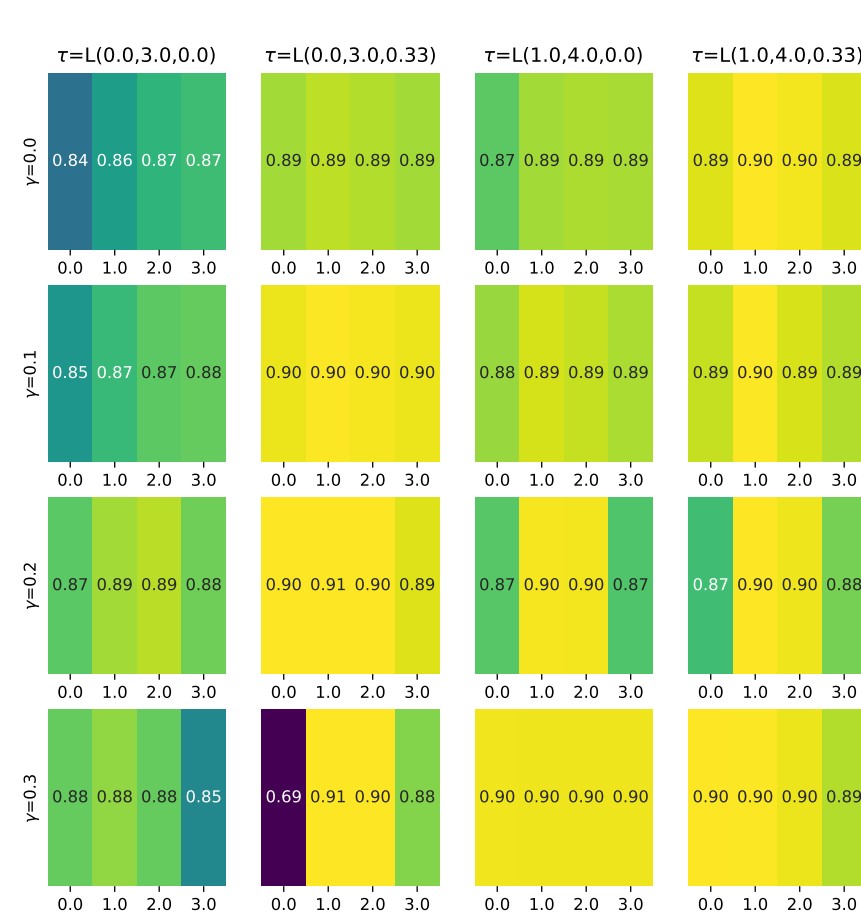

Figure 7: AUROC for 16 models trained with VS loss + LCT ($\lambda = \tau$). Each model was evaluated with 5 different hyperparameters (represented by the 5 small boxes inside of each large box). The models trained with $\gamma = 0.3$ and $\tau$ drawn from Linear(0.0, 3.0, 0.33) and evaluated with $\gamma = 0$, $\tau = 1$ give the best AUROC=0.908.

## G ADDITIONAL RESULTS

Tables 13 to 17 contains results analogous to Table 3 for other metrics. In other words, we train and evaluate over the hyperparameters in Table 2 and report the best values for each method. For each dataset, we bold the value corresponding to the best method and underline the value of the better method between the baseline and LCT.

Table 13: Average precision for various datasets and class-imbalance methods with and without LCT.

| Method | CIFAR-10/100 | | | | SIIM-ISIC Melanoma | | | APTOS | |
| | Auto Deer | Auto Truck | Animal Vehicle | House-hold | $\beta$=10 | $\beta$=100 | $\beta$=200 | $\beta$=100 | $\beta$=200 |
|---|---|---|---|---|---|---|---|---|---|
| Focal | 0.987 | 0.937 | 0.979 | 0.731 | 0.76 | 0.602 | 0.545 | 0.987 | **0.986** |
| Focal + LCT | 0.987 | 0.936 | 0.978 | **0.744** | 0.749 | 0.613 | 0.564 | **0.989** | **0.986** |
| VS | 0.983 | 0.929 | 0.979 | 0.738 | **0.763** | 0.614 | 0.550 | 0.983 | 0.983 |
| VS + LCT | 0.986 | 0.938 | 0.98 | 0.730 | 0.762 | **0.638** | **0.595** | 0.984 | 0.985 |
| VS + SAM | 0.987 | 0.931 | **0.985** | 0.741 | 0.682 | 0.521 | 0.507 | 0.644 | 0.623 |
| VS + SAM + LCT | **0.99** | **0.942** | 0.984 | 0.733 | 0.502 | 0.209 | 0.278 | 0.715 | 0.624 |

Table 14: Brier score for various datasets and training methods with and without LCT. **Lower values are better.**

| Method | CIFAR-10/100 | | | | SIIM-ISIC Melanoma | | | APTOS | |
| | Auto Deer | Auto Truck | Animal Vehicle | House-hold | $\beta$=10 | $\beta$=100 | $\beta$=200 | $\beta$=100 | $\beta$=200 |
|---|---|---|---|---|---|---|---|---|---|
| Focal | 0.122 | 0.256 | 0.103 | 0.397 | **0.036** | 0.047 | 0.052 | 0.140 | 0.192 |
| Focal + LCT | 0.113 | 0.23 | 0.094 | 0.283 | 0.037 | **0.046** | **0.05** | 0.117 | 0.177 |
| VS | 0.057 | 0.137 | 0.059 | 0.282 | 0.041 | 0.054 | 0.062 | **0.063** | **0.071** |
| VS + LCT | 0.049 | 0.164 | 0.047 | 0.361 | **0.036** | 0.048 | 0.053 | 0.115 | 0.114 |
| VS + SAM | 0.044 | **0.119** | 0.049 | 0.303 | 0.043 | 0.073 | 0.075 | 0.269 | 0.263 |
| VS + SAM + LCT | **0.038** | 0.134 | **0.042** | **0.278** | 0.063 | 0.074 | 0.076 | 0.248 | 0.248 |

Table 15: $F_1$ score for various datasets and class-imbalance methods with and without LCT. For each model, we find the maximum $F_1$ score over all possible thresholds.

| Method | CIFAR-10/100 | | | | SIIM-ISIC Melanoma | | | APTOS | |
| | Auto Deer | Auto Truck | Animal Vehicle | House-hold | $\beta$=10 | $\beta$=100 | $\beta$=200 | $\beta$=100 | $\beta$=200 |
|---|---|---|---|---|---|---|---|---|---|
| Focal | 0.948 | 0.854 | 0.925 | 0.685 | 0.691 | 0.558 | 0.514 | 0.963 | 0.948 |
| Focal + LCT | 0.949 | 0.858 | 0.922 | 0.686 | 0.679 | 0.562 | 0.532 | **0.964** | 0.951 |
| VS | 0.945 | 0.848 | 0.927 | **0.689** | **0.695** | 0.566 | 0.513 | 0.949 | 0.943 |
| VS + LCT | 0.947 | 0.86 | 0.929 | 0.685 | 0.689 | **0.582** | **0.544** | 0.958 | **0.953** |
| VS + SAM | 0.949 | 0.851 | **0.938** | 0.685 | 0.611 | 0.499 | 0.482 | 0.784 | 0.79 |
| VS + SAM + LCT | **0.958** | **0.866** | 0.934 | 0.685 | 0.577 | 0.455 | 0.432 | 0.730 | 0.692 |

Table 16: Balanced accuracy for various datasets and class-imbalance methods with and without LCT. For each model, we find the maximum balanced accuracy over all possible thresholds.

| Method | CIFAR-10/100 | | | | SIIM-ISIC Melanoma | | | APTOS | |
| | Auto Deer | Auto Truck | Animal Vehicle | House-hold | $\beta$=10 | $\beta$=100 | $\beta$=200 | $\beta$=100 | $\beta$=200 |
|---|---|---|---|---|---|---|---|---|---|
| Focal | 0.949 | 0.857 | 0.938 | 0.660 | 0.879 | 0.826 | 0.811 | 0.962 | 0.947 |
| Focal + LCT | 0.949 | 0.86 | 0.934 | 0.671 | 0.879 | 0.828 | 0.819 | **0.963** | 0.947 |
| VS | 0.946 | 0.850 | 0.938 | 0.664 | 0.883 | 0.819 | 0.799 | 0.949 | 0.943 |
| VS + LCT | 0.947 | 0.862 | 0.94 | 0.662 | **0.885** | **0.832** | **0.827** | 0.958 | **0.952** |
| VS + SAM | 0.950 | 0.853 | **0.948** | **0.675** | 0.862 | 0.807 | 0.806 | 0.622 | 0.625 |
| VS + SAM + LCT | **0.958** | **0.868** | 0.945 | 0.671 | 0.808 | 0.640 | 0.764 | 0.684 | 0.608 |

Table 17: Best precision when recall≥0.99 for various datasets and class-imbalance methods with and without LCT. For each dataset, we bold the value of the best method and underline the better value for each method with and without LCT.

| Method | CIFAR-10/100 | | | | SIIM-ISIC Melanoma | | | APTOS | |
| | Auto Deer | Auto Truck | Animal Vehicle | House-hold | $\beta$=10 | $\beta$=100 | $\beta$=200 | $\beta$=100 | $\beta$=200 |
|---|---|---|---|---|---|---|---|---|---|
| Focal | 0.746 | 0.582 | 0.641 | **0.511** | 0.155 | 0.107 | 0.104 | 0.746 | 0.741 |
| Focal + LCT | 0.75 | 0.584 | 0.644 | 0.506 | 0.156 | 0.111 | 0.118 | **0.767** | 0.746 |
| VS | 0.697 | 0.557 | 0.634 | 0.509 | 0.148 | 0.118 | 0.104 | 0.675 | 0.725 |
| VS + LCT | 0.725 | 0.585 | 0.647 | 0.506 | **0.158** | 0.125 | 0.126 | 0.733 | **0.761** |
| VS + SAM | 0.737 | 0.568 | **0.721** | 0.507 | 0.149 | **0.132** | **0.132** | 0.513 | 0.518 |
| VS + SAM + LCT | **0.79** | **0.591** | 0.713 | 0.504 | 0.124 | 0.098 | 0.110 | 0.512 | 0.510 |

## G.1 ADAPTABILITY OF FOCAL LOSS MODELS

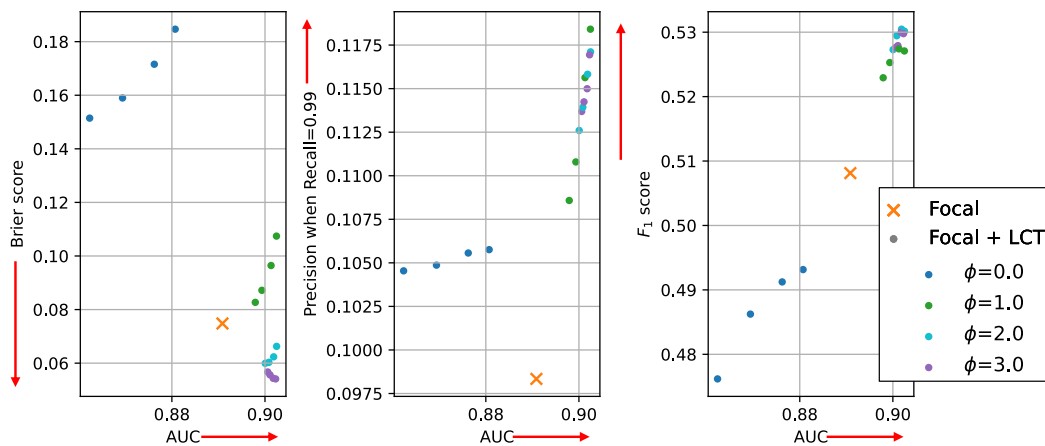

Figure 8: Results obtained from training one Focal loss model and one Focal + LCT model on the SIIM-ISIC Melanoma dataset with $\beta = 200$. For both methods, we show results for the model that obtains the highest AUC. For the LCT model, we evaluate with the 20 different $\lambda$ values from Table 2 and color the dots by the value of $\tau$. We compare AUC and Brier score (left), precision when recall is $\geq 0.99$ (center) and $F_1$ score (right). **One LCT model can be adapted *after* training to optimize a variety of metrics.**

## H FEATURE-WISE LINEAR MODULATION (FiLM)

Feature-wise Linear Modulation (FiLM) layers Perez et al. (2018) are layers that are added to a LCT model to condition it on $\boldsymbol{\lambda}$. These layers are small neural networks that take $\boldsymbol{\lambda}$ as input and output a $\boldsymbol{\mu}$ and $\boldsymbol{\sigma}$ that is used to modulate the activations within a network channel-wise. For example, suppose a convolutional layer in a network has activations $\boldsymbol{f}$ of size $W \times H \times C$. In Loss Conditional Training (LCT), these activations are augmented by $\boldsymbol{\mu}$ and $\boldsymbol{\sigma}$ channel-wise as follows,

$$\tilde{\boldsymbol{f}} = \boldsymbol{\sigma}\mathbf{f} + \boldsymbol{\mu} \tag{34}$$

where both $\boldsymbol{\mu}$ and $\boldsymbol{\sigma}$ are vectors of size $C$.

Figure 9 shows a diagram of FiLM layers being applied to a small example network. Normally, without FiLM layers, the equation to calculate the features maps is

$$\mathbf{x}^{(1)} = \rho(W^{(1)}\mathbf{x}^{(0)} + \mathbf{b}^{(1)}), \tag{35}$$

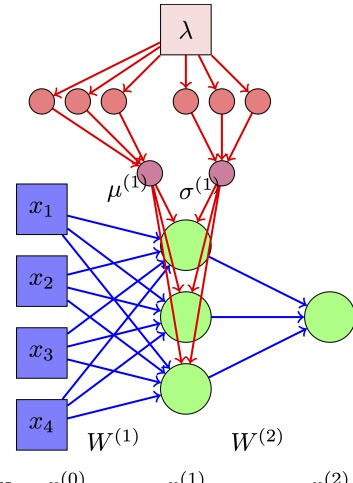

Figure 9: Cartoon of FiLM applied to a multi-layer perceptron.

where $\rho$ is the activation function (*e.g.*, ReLU), $W$ are the weights and $b$ are the biases in the layer. However, with FiLM layers, the weighted sum is first subject to augmentation with $\boldsymbol{\mu}$ and $\boldsymbol{\sigma}$. Thus, the new calculation is

$$\mathbf{x}^{(1)} = \rho(\boldsymbol{\sigma}^{(1)}(W^{(1)}\mathbf{x}^{(0)} + \mathbf{b}^{(1)})) + \boldsymbol{\mu}^{(1)}. \tag{36}$$

## I  STANDARD DEVIATIONS

For clarity, we do not include the standard deviation of the results over the three random seeds in the main body. We instead include these values in Tables 18 and 19 of this section.

Table 18: An expanded version of Table 3 for the CIFAR datasets that includes both the mean and standard deviation values.

| Method | CIFAR-10/100 | | | |
|---|---|---|---|---|
| | Auto Deer | Auto Truck | Animal Vehicle | House- hold |
| Focal | $0.985 \pm 0.004$ | $0.929 \pm 0.002$ | $0.982 \pm 0.002$ | $0.699 \pm 0.016$ |
| Focal + LCT | $0.985 \pm 0.002$ | $0.927 \pm 0.012$ | $0.981 \pm 0.001$ | $0.714 \pm 0.024$ |
| VS | $0.980 \pm 0.005$ | $0.918 \pm 0.018$ | $0.981 \pm 0.001$ | $0.710 \pm 0.004$ |
| VS + LCT | $0.983 \pm 0.002$ | $0.930 \pm 0.011$ | $0.983 \pm 0.001$ | $0.705 \pm 0.011$ |
| VS + SAM | $0.985 \pm 0.002$ | $0.923 \pm 0.005$ | $0.988 \pm 0.001$ | $0.709 \pm 0.010$ |
| VS + SAM + LCT | $0.988 \pm 0.002$ | $0.934 \pm 0.002$ | $0.987 \pm 0.000$ | $0.708 \pm 0.011$ |

Table 19: An expanded version of Table 3 for the SIIM-ISIC Melanoma and APTOS Diabetic Retinopathy datasets that includes both the mean and standard deviation values.

| Method | SIIM-ISIC Melanoma | | | APTOS Diabetic Retinopathy | |
| --- | --- | --- | --- | --- | --- |
| | $\beta$=10 | $\beta$=100 | $\beta$=200 | $\beta$=100 | $\beta$=200 |
| Focal | $0.951 \pm 0.001$ | $0.905 \pm 0.006$ | $0.891 \pm 0.002$ | $0.985 \pm 0.005$ | $0.982 \pm 0.004$ |
| Focal + LCT | $0.951 \pm 0.001$ | $0.910 \pm 0.007$ | $0.902 \pm 0.002$ | $0.987 \pm 0.001$ | $0.984 \pm 0.000$ |
| VS | $0.954 \pm 0.001$ | $0.905 \pm 0.002$ | $0.884 \pm 0.007$ | $0.980 \pm 0.007$ | $0.980 \pm 0.003$ |
| VS + LCT | $0.954 \pm 0.002$ | $0.914 \pm 0.001$ | $0.911 \pm 0.001$ | $0.983 \pm 0.002$ | $0.984 \pm 0.003$ |
| VS + SAM | $0.938 \pm 0.000$ | $0.895 \pm 0.002$ | $0.892 \pm 0.001$ | $0.613 \pm 0.134$ | $0.581 \pm 0.286$ |
| VS + SAM + LCT | $0.893 \pm 0.006$ | $0.650 \pm 0.198$ | $0.831 \pm 0.034$ | $0.715 \pm 0.058$ | $0.622 \pm 0.096$ |

