# OpenReview forum: "Training Over a Distribution of Hyperparameters for Enhanced Performance and Adaptability on Imbalanced Classification"
_ICLR.cc/2025/Conference — Submitted to ICLR 2025_

### Official Review · Reviewer_8q4P · 2024-11-02

**Soundness:** 3
**Presentation:** 3
**Contribution:** 3
**Rating:** 6
**Confidence:** 5

**Summary:**

The authors propose a new approach to training binary classification models in severely imbalanced situations, using loss-conditioned training (LCT) to train a single model on a hyperparameter distribution, which consistently improves ROC curves and various other metrics. In addition, training models using the proposed approach are more efficient because some hyperparameter tuning can be done after training to suit individual needs without having to retrain from scratch. The proposed algorithm achieves good performance on both natural and medical images.

**Strengths:**

The research addresses a significant topic in the field by proposing a method to solve the imbalance problem in binary classification tasks. The proposed approach effectively tackles this challenge, as evidenced by comprehensive and valid experiments conducted across multiple datasets. The method demonstrates promising results, outperforming a large number of comparison methods. Overall, the study offers a valuable contribution by providing a robust solution to a prevalent issue, supported by thorough experimentation and strong comparative performance.

**Weaknesses:**

The authors propose training the model at multiple hyperparameter values, aiming to approximate the best performance of multiple models trained using a single loss function. However, this approach may be unfair to other baseline methods. It is unclear how the hyperparameters for the baseline methods were selected in the experiments. Providing details on this selection process would help ensure a fair comparison.

The novelty of the proposed method is not very clear. While the authors demonstrate the impact of optimal hyperparameters on model performance through a series of intuitive experiments, it remains uncertain whether the performance gains are simply due to increased randomness during training. Clarification on this point would strengthen the contribution.

Additionally, it appears that the LCT algorithm is only applicable to binary classification tasks. In real-world scenarios, multi-class classification is more common, and binary classification may be too simplistic for addressing imbalanced problems. The authors should consider extending their method to multi-class classification to verify its effectiveness in more complex settings.

The comparison methods used in the experiments seem to be somewhat outdated, such as Focal Loss and VS Loss. Including comparisons with more advanced algorithms like Gaussian Clouded Logit (GCL) Loss [1] and Dual Focal Loss [2] would enhance the study’s relevance and rigor.

The performance improvements reported seem negligible, and it is not clear whether they are attributable to the hyperparameter settings. Notably, the proposed LCT method appears to improve VS Loss more significantly, but the improvement over Focal Loss is minimal. An explanation for this discrepancy would be valuable.

Since the models’ results are based on multiple runs, performing statistical tests to compare the performance of different algorithms and reporting the standard deviations would provide more robust evidence of the method’s effectiveness.

Finally, information about the computational requirements for training the proposed method and its algorithmic complexity would be beneficial. This would help assess the practicality of implementing the method in real-world applications.

References:
[1] Li, Mengke, Yiu-Ming Cheung, and Yang Lu. “Long-tailed Visual Recognition via Gaussian Clouded Logit Adjustment.” Proceedings of the IEEE/CVF Conference on Computer Vision and Pattern Recognition. 2022.
[2] Tao, Linwei, Minjing Dong, and Chang Xu. “Dual Focal Loss for Calibration.” International Conference on Machine Learning. PMLR, 2023.

**Questions:**

Please refer to the weakness section.

---

> ### Author Response · Authors · 2024-11-13
> **Inconsistent review- wrong paper?**
>
> Dear Area Chairs,
>
> We believe that this review was intended for a different paper because the review contains discussion of "Informative Data Selection (IDS)" and "Generative Data Augmentation (GDA)", which are not related to our work.
>
> Could you please re-evaluate the review and re-link it to the correct paper? Additionally, could you please check if there is another review intended for our paper?
>
> Thank you.

---

> > ### Comment · Area_Chair_1KGD · 2024-11-23
> >
> > Thank you for letting us know. We are looking into this.
> >
> > Best regards,
> >  - AC.

---

> > > ### Comment · Reviewer_8q4P · 2024-12-02
> > > **Review fixed**
> > >
> > > Dear all,
> > >
> > > Apologies for sending the wrong review; the correct review has been updated. Given that the deadline is approaching, I will omit the Questions section. Overall I give a positive recommendation for the acceptance of this work.

---

### Official Review · Reviewer_DgPZ · 2024-11-03

**Soundness:** 3
**Presentation:** 3
**Contribution:** 3
**Rating:** 6
**Confidence:** 3

**Summary:**

The paper extends the Loss Conditional Training framework to address class imbalance issues, based on the observation that varying hyperparameters in the loss function results in different performance across evaluation metrics. The proposed method shows good performance and desirable properties on several datasets.

**Strengths:**

The paper extends the Loss Conditional Training framework to binary classification with class-imbalance setting. The extension is simple but seems useful.

**Weaknesses:**

1. It is unclear how the distribution of lambda during training and the lambda values to use in inference are chosen. More discussion is needed.

2. The proposed method seems a simple extension of existing method (Dosovitskiy & Djolonga, 2020).

**Questions:**

1. Why does the paper focus solely on binary classification? Could the proposed method be extended to multi-class classification? What about class-balanced scenario?

2. The paper uses Focal Loss and VS Loss as examples of successful applications of the LCT framework. Why were only these two loss functions chosen? Are there more generalizable insights on which types of losses are or are not applicable?

4. Have the authors considered alternative methods for conditioning the model on lambda (lines 248-253)?

---

> ### Author Response · Authors · 2024-11-20
> **Response to reviewer DgPZ**
>
> Thank you for your thoughtful review of work. We are glad that you found our work to have strong potential. Additionally, we appreciate your comments and questions as it gives us the opportunity to improve the paper. We responded to some of your concerns in the general response and others inline below. We hope that these answers help clarify our work.
>
> ---
>
> *The paper uses Focal Loss and VS Loss as examples of successful applications of the LCT framework. Why were only these two loss functions chosen? Are there more generalizable insights on which types of losses are or are not applicable?*
>
> We chose to use VS loss because it is a generalization of many methods proposed for imbalanced classification. Additionally, Focal loss is an alternative loss that is effective for addressing imbalance and a popular choice in practice. While simple, these two loss functions encompass many loss functions used in the literature and provide a strong foundation for demonstrating the potential of training over a distribution of loss-function-hyperparameters, instead of a single value. We expect that our method would also improve other loss functions that have hyperparameters which control the tradeoff between accuracy on head and tail classes.
>
> ---
>
> *Have the authors considered alternative methods for conditioning the model on lambda (lines 248-253)?*
>
> We have not considered alternative methods for conditioning the model on lambda, but agree that this is an interesting suggestion. We used FiLM layers because these are used in the original LCT formulation.

---

> ### Comment · Reviewer_DgPZ · 2024-11-25
>
> I thank the authors for their effort preparing the rebuttal. I have read the authors' responses as well as comments from other reviewers. The authors' response partially address my concerns (i.e., Discussion on choosing Lambda). But the discussion on more general insight about the method is still unsatisfactory (when the proposed method will and will not work).
>
> Overall i will keep my original rating.

---

### Official Review · Reviewer_KA6n · 2024-11-04

**Soundness:** 2
**Presentation:** 2
**Contribution:** 2
**Rating:** 5
**Confidence:** 2

**Summary:**

This paper focuses on imbalanced classification. Authors focus on the hyper-parameter choices of imbalanced loss function and find no optimal solution for existing approach. Instead, the proposed method treating these hyper-parameters as a distribution and sample from it, combine with original data sample input as an additional input for training.

**Strengths:**

The idea of hyper-parameter sensitivity in imbalanced learning is new.

**Weaknesses:**

- How is SAM trained with LCT? Since $\lambda$ is associated with each data sample, but SAM is an optimizer?
- How is $P_\Lambda$ calculated? Is it a small network or it is just a pre-defined distribution?
- Line 248-253 is vague in details. How is FiLM compute $\mu$ and $\sigma$? Does it just take one $\lambda$ in and output 2 values? Because without any constraints, these two values are directly applied on activation for linear transformation.

**Questions:**

- I am confused by LCT training part. My understanding is that LCT will point out the best hyper-parameter after training, and in inference we just use the best combination. But Sec. 4.3 claim we still need to evaluate with multiple values $\lambda$. So LCT is more similar to meta-learning, through its training paradigm, the model can generalize better?
- Given there is an underlying distribution over these hyper-parameters, could a VAE approach work as well?

---

> ### Author Response · Authors · 2024-11-20
> **Response to reviewer KA6n**
>
> Thank you for your thoughtful review of work. We are glad that you found our work to be novel. Additionally, we appreciate your questions as it gives us the opportunity to clarify our work and improve the paper. We responded to some of your concerns in the general response and others inline below. We hope that these clarifications address your concerns.
>
> ---
>
> *How is SAM trained with LCT? Since $\lambda$ is associated with each data sample, but SAM is an optimizer?*
>
> We can train a model with both LCT and SAM because they affect different parts of the training procedure. As you correctly stated, when training with LCT, a $\lambda$ value is associated with each data sample. This value affects (a) the model’s inputs and (b) the calculation of the batch’s loss. After calculating the batch’s loss in the forward pass, we use an optimization method, like Stochastic Gradient Descent (SGD), to determine how to update the model’s weights to minimize that loss. Sharpness Aware Minimization (SAM) is simply an alternative to SGD and can be easily combined with LCT. Thank you for bringing this up, we have clarified this in Section 3.4 of the revised paper as well.
> How is $P_\Lambda$ calculated? Is it a small network or it is just a pre-defined distribution?
> It is a pre-defined distribution. We choose a distribution that includes reasonable values for each of the hyperparameters. We discuss the distributions that we use for $P_\Lambda$ in Section 4.3 and Table 2.
>
> ---
>
> *Given there is an underlying distribution over these hyper-parameters, could a VAE approach work as well?*
>
> We agree that there is potential for methods besides LCT to be used to train one model over a distribution of hyperparameters. If you could elaborate on this suggestion, we’d be interested in discussing more!

---

> > ### Comment · Reviewer_TQCe · 2024-11-26
> >
> > Thank the authors for their effort in preparing the feedback. Some of my concerns have been well addressed, such as the novelty of the  proposed method and details about FiLM. But I still have concerns on the clarity about the setting of hyper-parameters during training and testing and the explanation to some experimental results. Hence, I tend to keep my original score.

---

### Official Review · Reviewer_yRkB · 2024-11-04

**Soundness:** 2
**Presentation:** 3
**Contribution:** 2
**Rating:** 3
**Confidence:** 2

**Summary:**

The paper presents a method for training models on imbalanced datasets by using a distribution over loss hyperparameter values instead of a single value. The hypothesis is that by conditioning the model on different values of hyper-parameters, the model leads to different functions and thus, using a distribution of hyper-parameters leads to a more robust model. The method is evaluated on CIFAR and APTOS datasets.

**Strengths:**

* The paper is well-written, and the motivation is clear.

* The method is well-motivated and learning on imbalanced dataset is important.

**Weaknesses:**

* Only binary classification is studied in the paper. It is not clear if the proposed method will work on multi-class classification or unsupervised learning.
* The datasets used in the experiments are very small. Experiments on more large-scale datasets are required (e.g. ImageNet).
*Only focal loss and VS loss are used as baselines. More extensive comparison against Sota methods for learning imbalanced datasets and long-tail learning.
* Hyper-parameter search space for focal loss and VS loss is very small, making the comparison somewhat unfair.
* It is not clear how pdf for sampling $\lambda$ is defined in section 4.3.
* Authors should run the method on different kinds of learning problems in different domains.
* Improvement on CIFAR and SIIM-ISIC datasets ARE very minor.
* Standard deviation is not mentioned in the result table. How many different seeds were used for each section?

**Questions:**

Why was the model trained for 500 epochs on cifar (section 5.1)? That's excessive for CIFAR.

Why do you need gradient clipping? Is the loss unstable?

---

> ### Author Response · Authors · 2024-11-20
> **Response to reviewer yRkB**
>
> Thank you for your thoughtful review of work. We are glad that you agree this is an important area of work and think that our method is well-motivated. We responded to some of your questions and concerns in the general response and others inline below. We hope that after reading these responses you will have a better understanding of our work.
>
> ---
>
> *Only binary classification is studied in the paper. It is not clear if the proposed method will work on multi-class classification or unsupervised learning.*
>
> We chose to focus on binary problems for two reasons. First, despite the fact that many motivational examples in class imbalance are on binary problems, much of the state-of-the-art work in class imbalance has focused on longtailed multi-class problems. Thus, we wanted to study how these SOTA methods do or do not translate to binary problems. Second, studying binary classification problems allows us to use more interpretable evaluation metrics, like Precision-Recall curves, instead of aggregated metrics, like balanced accuracy. This helps us understand the mechanisms and limitations behind current work and describe the advantages of our proposed method. We fully expect that this method will work on multi-class classification and it should be easy to extend our method to this case. As for unsupervised learning, LCT has already been used in unsupervised problems, like Variational Autoencoders, and has shown that it can approximate several fixed-rate models for these applications, but at the cost of some performance degradation. Our contribution is showing that training a single model over a distribution of hyperparameters can improve the performance of models trained on datasets with class-imbalance, which inherently have labels.
>
> ---
>
> *The datasets used in the experiments are very small. Experiments on more large-scale datasets are required (e.g. ImageNet).*
>
> Respectfully, we disagree with this statement. One of the datasets we use (SIIM-ISIC Melanoma) has more than 45,000 training samples (please see table 12 in our submission for the number of samples each of our datasets has). Additionally, we described our rationale for focusing on binary datasets, instead of multi-class datasets like ImageNet, in the previous answer.
>
> ---
> *Hyper-parameter search space for focal loss and VS loss is very small, making the comparison somewhat unfair.*
>
> Again we respectfully disagree with this statement. We chose wide hyperparameter ranges that encompass the best-performing models in terms of AUC. Using a wider range of hyperparameters would only lead to training more models with poor performance.
>
> ---
>
> *It is not clear how pdf for sampling $\lambda$  is defined in section 4.3*
>
> We use a linear probability distribution function (pdf), which is a general form of probability distribution functions like uniform and triangular distributions. Our submission included a detailed explanation of this in Appendix D. Please let us know if you have more specific questions about this distribution or how to sample from it.
>
> ---
>
> *Standard deviation is not mentioned in the result table. How many different seeds were used for each section?*
>
> As stated in line 350 of our submission, we use 3 random seeds for each set of hyperparameters. We left standard deviation off of the table in the main body for clarity, but added those results to Appendix I.
>
> ---
>
> *Why was the model trained for 500 epochs on cifar (section 5.1)? That's excessive for CIFAR. Why do you need gradient clipping? Is the loss unstable?*
>
> We train for 500 epochs because LCT models take longer to converge since they are training over a distribution of hyperparameters, instead of a single value. We found that training the baseline models longer does not hinder their performance and do this for consistency. Additionally, we use gradient clipping because the randomness added by LCT can cause spikes in the loss, especially early in training.

---

> > ### Comment · Reviewer_yRkB · 2024-11-25
> > **Reply to the rebuttal**
> >
> > Thank you for the reply. After going through other reviews, I think the paper lacks clarity, particularly in how $lambda$ is being sampled.
> >
> > I am fully convinced with the rationale for studying binary classification problems exclusively; this limits the practical application of methods.
> >
> > While a large dataset is difficult to define, contemporary deep learning papers do not consider 45,000 training samples to be a large dataset.
> >
> > I would like to keep my original rating! I suggest addressing the limitation of the work pointed out be reviewers in the next round of submission.

---

### Official Review · Reviewer_TQCe · 2024-11-04

**Soundness:** 4
**Presentation:** 4
**Contribution:** 2
**Rating:** 5
**Confidence:** 4

**Summary:**

This paper is targeted at addressing the imbalanced image classification task. It reveals that the performance of existing methods trained using fixed hyper-parameters is sensitive to hyper-parameters, namely the highest precision is achieved with different hyper-parameter values at different recall. Targeted at learning a single model which can achieve consistently high performance across recall, it introduces the loss conditional training strategy which is originally proposed for image compression into image classification. This enables the hyper-parameters to flexibly tuned during inference. Extensive experiments on a few binary image classification datasets demonstrate the proposed method improves baseline methods focal loss and vector scaling loss consistently.

**Strengths:**

1. The paper presents a novel integration of loss conditional training (LCT) within the context of imbalanced image classification. While LCT itself is an existing technique, its application in this specific domain brings new insights, particularly in handling varying hyper-parameters to improve model adaptability and performance on diverse datasets.
2. The research demonstrates rigorous experimental validation. By applying the proposed method across multiple binary image classification datasets, the paper effectively highlights its potential in improving baseline performances. The objective function is thoughtfully designed, focusing on minimizing classification loss over a hyper-parameter distribution, which is both innovative and practical.
3. The paper is generally well-structured and presents its contributions clearly. The explanation of the objective function and its rationale is straightforward, aiding readers in understanding the methodology.
4. Given the widespread challenges posed by imbalanced datasets in image classification, the method’s capability to flexibly adapt post-training, as well as its empirical success on standard benchmarks, underscores its relevance and significance in the field.

**Weaknesses:**

1. While the application of LCT to imbalanced classification is a reasonable extension, the core technique—LCT—is not novel. The contribution could be viewed as incremental, leveraging existing methodologies without substantial innovation.
2. The choice of baselines seems outdated. To substantiate the method's efficacy, comparisons with more recent state-of-the-art techniques should be included：
SuperDisco: Super-class discovery improves visual recognition for the long-tail, cvpr2023
Balanced product of calibrated experts for long-tailed recognition, cvpr2023
Constructing balance from imbalance for long-tailed image recognition, eccv2022
3. The paper lacks sufficient details on the FiLM (Feature-wise Linear Modulation) module, which is critical for reproducibility. A clearer explanation and inclusion of implementation specifics are necessary.
4. The requirement for hyper-parameter input during inference could pose practical challenges. Providing a guideline for optimal hyper-parameter settings would be beneficial for practitioners and researchers aiming to apply this method.
5. The method's inability to outperform VS-SAM in several settings is concerning. This raises questions about the generalizability and robustness of the approach. Furthermore, the observation from Table 4, where LCT without FiLM underperforms compared to the baseline, warrants a deeper analysis to understand the underlying reasons.

**Questions:**

1. Could the authors clarify how their approach extends beyond a straightforward application of LCT? Highlighting specific innovations or modifications would help in assessing the paper’s originality.
2. Could the authors provide more details on the FiLM implementation?
3. The method relies on hyper-parameter inputs during inference. What guidelines or heuristics do the authors recommend for setting these parameters effectively?
4. The results in Table 4 indicate that without FiLM, LCT performs worse than the baseline. Could the authors provide an analysis of why FiLM is critical and why its absence degrades performance? Similarly, why does the method not consistently outperform VS-SAM?

---

> ### Author Response · Authors · 2024-11-20
> **Response to reviewer TQCe**
>
> Thank you for your thoughtful and detailed review of work. We are glad that you found our work to be relevant and significant. Additionally, we appreciate your comments and questions as it gives us the opportunity to improve the paper. We responded to most of your concerns in the general response and the others inline below. We hope that these clarifications address your concerns.
>
> *The results in Table 4 indicate that without FiLM, LCT performs worse than the baseline. Could the authors provide an analysis of why FiLM is critical and why its absence degrades performance? Similarly, why does the method not consistently outperform VS-SAM?*
>
> Yes. The FiLM layers are critical because without them the models do not receive $\lambda$ as an additional input. Specifically, when we train with LCT without FiLM, we use $\lambda$ to control the loss function of each batch, but never feed this information to the model. Thus, from the model’s perspective, the loss value is partially random. This has the effect of hindering training since the model is learning random noise.
> The interaction between LCT and SAM is more complicated. While both LCT and SAM can improve the performance of baseline models, we observe that using them together produces inconsistent results. We find that when combining SAM with LCT, many models fail to converge and, for some, training completely fails due to exploding gradients. This suggests that there may be an underlying conflict between LCT and SAM. SAM introduces variability in the training process because it uses the gradient at an adversarial point, instead of the actual point. LCT introduces additional variability by drawing different hyperparameters with each batch and conditioning the model on this. This dual source of variability might result in the model moving in nonsensical directions on the loss landscape and reduce the model’s ability to converge to effective solutions. Studying the interaction between these two methods is an important area of future work.

---

### Author Response · Authors · 2024-11-20
**Global response 1**

We thank the reviewers for their valuable time and feedback. We are glad that the reviewers observed that “[t]he idea of hyper-parameter sensitivity in imbalanced learning is new” (KA6n) and agree that our proposed remedy is “innovative and practical” (TQCe), “well-motivated” (yRkB), and “simple but seems useful” (DgPZ). Additionally, they agreed that the empirical results show that this method “improve[s] model adaptability and performance on diverse datasets” (TQCe) and “shows good performance and desirable properties on several datasets.” (DgPZ). Finally, the reviewers agree that “learning on imbalanced dataset is important” (yRkB) and “[our] method’s capability to flexibly adapt post-training, as well as its empirical success on standard benchmarks, underscores its relevance and significance in the field.” (TQCe).

We provide one general response for questions that came up multiple times here and respond to each reviewer individually.

---

*Questions about how to choose $\lambda$ for inference (TQCe, KA6n, DgPZ).*

**Short answer**: we recommend using a validation set to choose the $\lambda$ value used at inference time.

**Long answer**: We thank the authors for bringing up this important nuance. While it is true that the inference $\lambda$ is a hyperparameter that requires a setting, this should not pose practical challenges because it can simply be done once right after training (any changes beyond this are completely optional). Specifically, we suggest that after training a model with LCT, the practitioner finds the optimal $\lambda$ for inference using a validation set and whatever their preferred metric(s) is(are). Note that this is a very efficient process because it only requires running inference on a validation set and not re-training the model. Thus, tuning an inference $\lambda$ hyperparameter takes a fraction of the time that tuning a loss-function hyperparameter does. After choosing this $\lambda$, the practitioner can simply use this value for all inferences. If, however, there is a change in their desired metric, they can repeat this process (tuning by running inference on the validation set) to find the new optimal $\lambda$ for that metric and adapt the model accordingly.

---

*Concerns that the method is a simple extension of LCT (TQCe, DgPZ).*

Thank you for bringing up this important nuance. As you correctly noted in your reviews, we use LCT, which is an existing technique, to train one model over a distribution of hyperparameters. **However, unlike past work that used LCT to approximate several existing models with one model, we use LCT to improve model performance.** LCT was proposed as a way to simplify training and reduce the number of similar models needed to perform the same task at different rates; in this context it suffered a small performance **penalty** in exchange for reduced computation and memory. We, on the other hand, find that in this domain (imbalanced classification), training over a distribution of hyperparameters can actually **improve** performance over baseline models. We propose using LCT to exploit the sensitivity of models to hyperparameters and train one more-general model. Additionally, we identify that different hyperparameters optimize for different points along a precision-recall curve and training one model over a distribution of hyperparameters is equivalent to optimizing over a wide region of the precision-recall curve, instead of optimizing for a single tradeoff on the curve.

---

> ### Author Response · Authors · 2024-11-20
> **Global response 2**
>
> *Requests for more explanation about the FiLM module (TQCe, KA6n).*
>
> We apologize for the lack of clarity about the Feature-wise Linear Modulation (FiLM) module used in the LCT models. We have added a section to the appendix (Section H) that explains these and includes a diagram to help the reader visualize this. Additionally, we provide implementation details in Section 5.1 under “model architecture.”
>
> *Choice of baselines (TQCe, KA6n, DgPZ).*
>
> We appreciate the reviewers’ suggestions for additional baseline methods. We would like to clarify that many of the State-of-the-Art methods are primarily designed for multi-class long-tailed recognition problems and many of the ideas proposed within them are not directly applicable to the binary classification setting considered in our work. For example, SuperDisco (CVPR 2023) introduces super-class discovery to learn the hierarchical relationship between classes in longtailed multi-class classification settings. Binary classification, by definition, does not involve such inter-class relationships, making this approach unsuitable for our problem. Similarly, Constructing balance from imbalance for long-tailed image recognition (ECCV 2022), uses Dynamic Label Space Adjustments (DLSA) to first separate head and tail classes to create a series of more balanced classification tasks for one longtailed multi-class classification task. Balanced product of calibrated experts for long-tailed recognition (BalPoE) is more transferable since it uses an ensemble of models, each trained with different logit adjustments. Each of these models is a special case of VS loss. We are actively working on comparing our models with BalPoE.
>
> VS loss is a generalization of many methods proposed for imbalanced classification, including the logit adjustments proposed by BalPoE. Focal loss is another commonly-used and effective method for addressing imbalance. While simple, these two loss functions encompass many loss functions used in the literature and provide a strong foundation for demonstrating the potential of training over a distribution of loss-function-hyperparameters, instead of a single value.
>
> We hope this clarifies our choice of baseline methods. We have added the suggested references to our related works section and agree that comparing these methods to ours in the multi-class case is important future work.

---

### Meta-Review · Area_Chair_1KGD · 2024-12-09

**Metareview:**

This paper was reviewed by 5 experts in the field and received 5, 3, 5, 6, 6 as the ratings. The reviewers agreed that the paper addresses the important problem of imbalanced data classification, the method is well-motivated, and the paper is well-written.

Reviewers yRkB and DgPZ raised a concern that the proposed method has been studied only on binary classification problems and it is not clear if the method will work on multi-class settings. The authors have mentioned that binary classification problem will enable the usage of more interpretable evaluation metrics, like precision-recall curves, instead of aggregated metrics, like balanced accuracy. However, this cannot be considered as a convincing argument, as many state-of-the-art methods in imbalanced data classification have focused on multi-class problems.

Concerns were also raised regarding the choice of baselines, where it was mentioned that a more extensive comparison against state-of-the-art baselines is necessary. While the authors have mentioned that many state-of-the-art baselines are designed for multi-class long-tailed recognition problems and they are not directly applicable to the binary classification setting, this cannot be considered as a convincing argument. Focusing only on binary classification problems limits the application scope of this method; it also becomes difficult to assess the merit of the framework without a thorough comparison against the state-of-the-art techniques.

Reviewers TQCe and DgPZ have mentioned that the proposed method is a simple extension of the Loss Conditional Training (LCT) method, which limits its novelty. The authors have clarified that  while past research has used LCT to approximate several existing models with one model, they have used LCT to improve model performance. However, this cannot be considered as a substantially novel contribution, given the standards of ICLR.

It was also mentioned that the proposed method has been validated only on small datasets. This issue was not addressed convincingly by the authors, as 45,000 training samples cannot be considered large-scale. Moreover, further discussion is necessary on how the parameter $\lambda$ is sampled.

We appreciate the authors' efforts in meticulously responding to each reviewer's comments. However, in light of the above discussions, we conclude that the paper may not be ready for an ICLR publication in its current form. While the paper clearly has merit, the decision is not to recommend acceptance. The authors are encouraged to consider the reviewers’ comments when revising the paper for submission elsewhere.

**Additional Comments On Reviewer Discussion:**

Please see my comments above.

---

### Decision · Program_Chairs · 2025-01-22

Reject